

**"Everything is scorched by the burning sun": Missionary perspectives and experiences of 19th and early 20th century droughts in semi-arid central Namibia**

Stefan Grab[1], Tizian Zumthurm[,2,3]

[1] School of Geography, Archaeology and Environmental Studies, University of the Witwatersrand, South Africa

[2] Institute of the History of Medicine, University of Bern, Switzerland

[3] Centre for African Studies, University of Basel, Switzerland

*Correspondence to*: Stefan Grab (Stefan.grab@wits.ac.za)

**Abstract.** Limited research has focussed on historical droughts during the pre-instrumental weather-recording period in semi-arid to arid human-inhabited environments. Here we describe the unique nature of droughts over semi-arid central Namibia (southern Africa) between 1850 and 1920. More particularly, our intention is to establish temporal shifts of influence and impact that historical droughts had on society and the environment during this period. This is achieved through scrutinizing documentary records sourced from a variety of archives and libraries. The primary source of information comes from missionary diaries, letters and reports. These missionaries were based at a variety of stations across the central Namibian region and thus collectively provide insight to sub-regional (or site specific) differences in hydro-meteorological conditions, and drought impacts and responses. Earliest instrumental rainfall records (1891-1913) from several missionary stations or settlements are used to quantify hydro-meteorological conditions and compare with documentary sources. The work demonstrates strong-sub-regional contrasts in drought conditions during some given drought events and the dire implications of failed rain seasons, the consequences of which lasted many months to

several years. The paper advocates that human experience and associated reporting of drought events depends strongly on social, environmental, spatial and societal developmental situations and perspectives. To this end, the reported experiences, impacts and responses to drought over this 70 year period portray both common and changeable attributes through time.

## 1. Introduction

Defining *drought* as a 'concept' or as an 'event' has received much discussion and debate, which seems ongoing (e.g. Agnew and Chappell, 1999; Mishra and Singh, 2010; Lloyd-Hughes 2014; Parry et al., 2016). In this special issue, Brázdil et al. (2019) explore various types and characteristics of drought that are relevant to both contemporary and historical contexts. These authors use the definition by Wilhite and Pulworty (2018) to define drought as 'a prolonged period of negative deviation in water balance compared to the climatological norm in a given area' (p1915). Although quantification of 'cimatological norms' during pre-instrumental periods is challenging, if at all possible, we broadly follow Wilhite and Pulworty's definition of drought for our current work. Today most water-requiring situations for agriculture, industry and human consumption etc, is to a large extent controlled through engineered water transfer schemes, water storage and water extraction. Hence, contemporary meteorological droughts may not necessarily culminate in agricultural or economic droughts owing to human-engineered interventions. Conversely, societal expansion with associated increasing extraction demands on river, lake and sub-surface water resources may induce ecological droughts that would otherwise not have occurred under given hydro-meteorological conditions. The nature of recent and contemporary droughts in its various contexts is thus becoming increasingly complex. For this reason, there is value in exploring drought contexts through a window of time when the natural-human environment was rapidly transformed into a more human-engineered environment (through colonial conquests). For instance, it may provide insight to how drought impacted past indigenous populations and the environment, in ways that may no longer apply today, such as water-resource contexts during periods of nomadic lifestyles.

Although drought is recognized as an environmental and climatic disaster (Mishra and Singh, 2010) which impacts many sectors such as agriculture, economy, human social dynamics, human health and ecosystems (Esfahanian et al., 2016), its influence may be highly variable

depending on its intensity and duration within particular climatic regimes. 'Drought' is
differentiated from 'aridity' where the former is considered a temporary phenomenon and the
latter a permanent one (Hisdal and Tallaksen, 2000). To this end, it may be a challenge on
perspective to differentiate between drought and aridity in semi-arid regions with a strong
bimodal rainfall distribution. Drought in such already water-stressed regions during 'normal
climatic conditions', may have far reaching effects and implications that are not applicable to
those of better watered regions such as for instance central Europe or most parts of North
America. Central Namibia is a semi-arid to arid region characterized by climatic extremes,
seasonal aridity and prolonged droughts (Grab and Zumthurm, 2018), and thus offers an ideal
spatial context to explore attributes of historical droughts in an already dry environment.

Most documentary-based southern African climate chronologies are focussed only on the 19th
century and end in 1899 or 1900 (e.g. Nash and Endfield, 2002, 2008; Kelso and Vogel 2007,
Grab and Nash, 2010; Nash and Grab, 2010; Nash et al., 2016, 2018), as was also the case with
that for central Namibia (Grab and Zumthurm 2018). However, given that the colonial period
with relatively poor instrumental weather records extended into the 20th century in many parts
of southern Africa, it is perhaps unfortunate that most studies have not extended their
chronologies into the 20th century. This is particularly so given that the early 20th century
experienced some severe droughts. While Grab and Zumthurm (2018) considered
climatological causes for 19th century wet-dry periods over central Namibia, the current paper
focuses on the broader context of historical droughts (consequences, perceptions, socio-
economic, socio-political, ecological) during the period 1850-1920. Extending previous work
to 1920 permits the placement of 19th century droughts in context with those during the early
20th century in central Namibia. Such a temporal extension is particularly valuable given rapid
societal change associated with technological and infrastructural advancements during the late
19th/early 20th centuries. Here we investigate how drought events are portrayed through textual
sources written by early European colonists (primarily missionaries) in what is today central
Namibia. Similar approaches have been taken to conceptualize climatic variability and
associated human responses in the adjoining semi-arid/arid regions of the Kalahari (e.g. Nash
and Endfield, 2002; Endfield and Nash, 2002) and Namaqualand (Kelso and Vogel, 2015).
This then provides us with an opportunity to establish similarities and differences in 19th C
drought-related circumstances and experiences through dryland regions of southern Africa.
More particularly, we aim to:1) outline the historic context of meteorological/hydrological
drought over central Namibia, 2) provide evidence for the (at times) relatively complex
geographic nature (spatial/temporal) of such droughts in the region, 3) summarize central
Namibian drought events between 1850 and 1920, and 4)  establish the temporal shifts of
influence and impact that historical droughts had on society and the environment during this
period, as portrayed in written documents. At this juncture, it is important to emphasise that
the perspectives, interpretations and views presented are entirely those expressed by European
colonists, and in particular from the spatial context of missionary stations. Regrettably, there
are few, if any, 19[th] century documents written from the perspectives of indigenous
communities, who may have had different views on 'drought' in central Namibia. Nonetheless,
documentary sources permit, to some extent, to sketch out some of the consequences and
responses to drought by the indigenous population living within relative proximity to mission
stations.

**2.  Data and Methods**

This paper is based on early documentary records from central Namibia, but also includes the
earliest instrumental rainfall records from various stations between 1891 and 1913.

Documentary sources used are the same as those described in detail by Grab and Zumthurm
(2018), and particularly those associated with the Rheinische Missionsgesellschaft [Mission
Society](RMS). The Society released annual reports describing conditions at each (or most) of
its mission stations and thus permits comparison across various sub-regions each year. Details
were less comprehensive in earlier years but as more mission stations were established through
the course of time, reporting became increasingly widespread and better informed (here we
refer the reader to Figure 3 in Grab and Zumthurm, 2018). Missionary Carl Hugo Hahn's
diaries (1850-1859) are an invaluable source of information for the earliest years. The
following are primary sources of documentary records used, especially to understand the
context of droughts as experienced and portrayed through German missionaries: ARRMS
(Annual Reports of the Rheinische Missionsgesellschaft- Archives of the Mission 21, Basel,
Switzerland), BRM (Berichte der Rheinischen Mission [Reports of the Rhenish Mission]),
sourced from the Archives of the Evangelical Lutheran Church, Windhoek, Namibia), and
station chronicles RMG (Rheinische Missionsgesellschaft [Rhenish Mission Society], sourced
from the Archives of the United Evangelical Mission [VEM], Wuppertal, Germany).
Prominent missionaries who spent many years in Namibia include: Carl Hugo Hahn (based at
Otjikango), Heinrich Kleinschmidt (based at Rehoboth), Franz Heinrich Vollmer (based at
Rehoboth and later Hoachanas), Johann Carl Böhm (based at Ameib and Rooibank), Johann
Jakob Irle (based at Okahandja and Otjosazu), Friedrich Wilhelm Viehe (various stations),
Johann Heidmann (based at Rehoboth), Philipp Diehl (based at Okahandja and Hoachanas),
and Peter Friedrich Bernsmann (based at Otjimbingue and Omburo). For later years (1894/5
onwards), annual reports, written by district officials and resident magistrates, are
exceptionally  valuable written sources of information as these summarise weather/climatic
conditions for various sub-regions each year, as also report on agriculture, grassland/grazing
conditions, disease, health, state of the environment etc. – these were sourced from the National
Archives of Namibia (NAN) in Windhoek.  A variety of other relevant documentary sources
were accessed through the Cape Archives Depot (CAD) at the Western Cape Provincial
Archives in Cape Town, and Evangelisch-Lutherische Kirche in Namibia (Evangelical
Lutherin Church in Namibia)(ELKIN). Several detailed travel logs/diaries from individuals
(e.g. C.J. Andersson, A. Eriksson, J. Chapman, A. Henker) were also consulted and recorded
at the various archives mentioned above, including also the William Cullen Library archives at
the University of the Witwatersrand.

We photographed and digitized the earliest available instrumental rainfall records (monthly
totals); these were sourced from the '*Mitteilungen aus den Deutschen Schutzgebieten*', Band
XXXII. The records cover the stations of Rehoboth (south), Windhoek (central highlands) and
Okahandja (northern highlands) for the period 1891-1913 (Figure 1). Additional station records
for the drier western region (Otjimbingue) and wetter eastern region (Gobabis) are also
included, covering the years 1899-1913 and 1897-1913 respectively. These records provide
valuable insight to seasonal and inter-annual rainfall variability during the late 19th/early 20th
centuries, as also spatial differences in given months, seasons and years. These are then used
to compare against the documentary records and to quantify the severity and duration of
drought/dry conditions.

Grab and Zumthurm (2018) provide methodological detail on how the documentary sources
were used to construct a 19th century climate chronology. This chronology was used in our
current work, in consultation with a re-evaluation of the documentary sources, to identify
periods of drought between 1850 and 1920.  The instrumental rainfall records assist to not only
identify, but also quantify drought events since ca 1891. The documents were further
scrutinized to establish attributes and consequences of these droughts (climatic, social-
responsive, social-environmental), in particular focusing on spatial and temporal contexts
(Table 1). A primary objective is to determine whether droughts may have had changing
impacts on society and the environment through time (i.e. 70 years of the study). Although in
less detail than what our study presents here, Kelso and Vogel (2015) also examined the
impacts of drought on livelihoods (resilience) in Namaqualand (to the south of our current
study area) through the 19th C. More recently, Pribyl et al. (2019) examined the role of drought
in agriarian crisis and social change over south-eastern Africa during the 1890s.

As is the case with most such studies, it is important to acknowledge potential data and
methodological limitations. In this case, it is important to recognize that the quantity and spatial
coverage of information was variable and more limited in earlier years than latter years, or
during years of war/severe conflict. To this end, some attributes associated with specific
droughts may have gone unreported. As already mentioned, the perspectives presented here are
Eurocentric (for reason of data availability) and from particular geographic settings (i.e.
stations located next to rivers or a 'permanent' water source) within the broader landscape
(space).

**3.   The historic central Namibian rainfall/drought context**

Mean annual rainfall across central Namibia (1891-1913) was highly variable, ranging from
384-413mm in the better watered central and eastern highland regions (Okahandja, Windhoek,
Gobabis), to 254mm in the southern region (Rehoboth) and 174mm in the western part
(Otjimbingue) (Figure 1). Inter-annual rainfall variability is higher (and thus less reliable) in
the drier regions (Grab and Zumthurm, 2018). Rainfall is strongly seasonal, with 95% falling
over the austral summer/autumn seasons (November-April). The long dry season (May-
October) rarely has rain of any consequence, and averages from as little as 8mm/pa at
Otjimbingue to 25mm/pa at Gobabis. Several months without any rainfall during the dry
season is thus the norm for central Namibia. This has important implications for when/where
the rain season has been considerably below average, as it places enormous stress, challenges
and consequences for surviving the long dry months. Vegetation patterns, human/animal
movements, and human economies during pre-colonial times were adapted to these semi-
arid/arid conditions across the region, with its annual cycle of brief summer rains and several
months of little to no rainfall (McCann, 1999).

Indigenous African inhabitants to central Namibia, before and during the 19th century, would
have been familiar with such seasonal climatic patterns and adapted their lives to best cope
with environmental conditions. People moved around with their livestock or planted and
harvested crops at specific localities and times of the year, thereby navigating the impacts of
extreme seasonal hydro-climatic variability or extreme climatic events.  While scholars have
identified typical hunter-gatherer, agropastoralist and pastoralist groups for precolonial central
Namibia (e.g. Gschwender, 1994/95), such distinctions were not unambiguous. Almost all
communities hunted regularly, farmed and gathered occasionally/episodically, and kept
varying numbers of sheep, goats, or cattle. Furthermore, such communities exchanged goods
amongst each other and traded with neighbouring groups and beyond (Wallace, 2011).
Consequently, political and economic dominance was tangible. In particular, much of central
Namibia's economy functioned through cattle, which was viewed to be the best option to store
wealth, as it was easily transferable. Combined with smart and shifting alliance-making, large
herds of cattle allowed its controller to enforce tribute-systems or to claim land and thus ensure
political dominance. Such a socio-economic system was, however, easily disrupted through a
variety   of   factors   such   as   drought,   conflict,   cattle   diseases   and   European
colonization/influence.  As also reported for other regions of southern Africa (e.g. Pribyl et al.,
2019), such an indigenous socio-economy gradually declined in significance as European
influences rapidly increased through the late 19th/early 20th centuries,

The establishment of permanent missionary and other European settlements in the region from
the mid 19th century onwards, altered local power dynamics, and brought about gradual change
to some aspects of societal lifestyles and the environment.  It was the missionaries' desire and
calling to attract local inhabitants towards permanent settlement at mission stations in order to
not only control and finally convert them, but also to teach them, among many other things,
western agricultural principles that they considered superior to those used locally. These
processes would help fulfil the colonial conquest. Consequently, this gradually changed the
'open indigenous agricultural economies' to more 'closed agricultural economies' (Ballard,
1986) which became increasingly dependent on local harvests, grazing and water resources,
and employment.  Inevitably, as will be demonstrated, this led to increased vulnerability and
social tensions during times of drought.  Given that the importance of cattle as a means of
subsistence and wealth  continued through the 19th and early 20th centuries, grazing conditions
were used as an important attribute to defining the severity of drought by local inhabitants

(European and indigenous). However, we acknowledge that factors such as locust invasions, livestock pressures (e.g. overgrazing) and fires would also have influenced grazing conditions. Hence, while climate (droughts) undoubtedly influenced social change, this always requires a critical assessment to avoid the trap of 'climate determinism' (see Hannaford et al., 2014).

Arguably the most significant and recurring extreme climatic event affecting central Namibia during the period 1850-1920 was drought. Given the region's strong bimodal rainfall pattern, Europeans writing from the area during earlier years of settlement, sometimes reported the occurrence of 'drought' during the dry season. However, as demonstrated, several months without rain during the dry season is 'normal' and thus does not constitute drought, but rather dry season aridity. It is important to recognize that those reflecting and reporting on the central Namibian environment and its climate were mostly German missionaries who would have been accustomed to a much cooler and wetter Germany. Although colonists would have arrived in semi-arid central Namibia with a likely central-northern European perspective on 'drought', any naivety concerning the local context would have changed as they became familiar with their new environs and interacted and learnt from local inhabitants and fellow missionaries who were familiar with the past and contemporary climate. For instance, after an initial four years in central Namibia, missionary Kleinschmidt reports from Rehoboth on 3 October 1846, that this is the '*worst*' time of year with respect to water availability and grazing (i.e. end of the long dry season). He further comments that there had only been limited rain during the last years and that grass recovery was only moderate (ARRMS, 1847, 145). Such comments suggest that while Kleinschmidt was familiar with the cyclic nature of annual rain and dry seasons, perhaps the assessment of there having been limited rain and moderate grass recovery is one of perspective, still in part influenced from his region of upbringing in modern day Lübbecke, Germany. Lübbecke has a sub-Atlantic maritime climate with all-year rainfall and thus grass remains relatively green throughout the year. To this end, and where possible, comments on weather, climate and the environment require careful scrutiny and comparison across various sources. In most cases written texts contain valuable contextual information (e.g. dryness/wetness of river channels, poor state of shrubs and trees, comments from older indigenous inhabitants etc) which helps verify claims of drought. In addition, several missionaries resided and travelled extensively in central Namibia for many years and in some instances decades (e.g. Viehe: 26yrs; Hahn: 30yrs; Heidmann: 39yrs; Bernsmann: 42yrs; Irle: 47yrs; Diehl: 51yrs), constantly interacting with local community members. In such cases, missionaries developed excellent knowledge of the local weather patterns and climate, and

were able to place contemporary climatic conditions in perspective, comparing situations with those experienced over many years prior. Two examples follow which place the severe droughts of 1902 and 1908 in perspective with the worst droughts recalled from the second half of the 19th century:

"*In the 31 years that missionary Heidmann was in Rehoboth, he had never experienced such a dry year as this*" [1902] (ARMS, 1902, 20). In addition, "*Missionary Irle, who had been in the region since 1869, could not remember the water table ever having been this low* [in 1902]" (ARMS, 1902, 29).

"*In the 34 years that missionary Dannert has been here* [Omaruru*], he can only recall the drought of 1879 being as severe as the one felt now* [1908]" (ELCIN, V.23.1, 351).

## 4. Results
### 4.1 Droughts in central Namibia (1850-1920)

Please also refer to the work by Grab and Zumthurm (2018) who describe relatively dry and very dry (drought) years over central Namibia between 1850 and 1900. Our current focus will only be on 'very dry' (drought) years; namely those of 1850-51, 1858-60, 1865-69, 1877-79, 1881-82, 1887-90, 1895-96, 1900-03, 1907-08, 1910-11 and 1912-13 (Figure 2). Figure 2 lists the number of times 'drought' is mentioned in documentary sources each year, and how this compares with the hydro-meteorological 19th C chronology produced by Grab and Zumthurm (2018). While the depicted results are impacted by documentary data availability and do not necessarily indicate drought severity, the intention with this figure is to provide a visual impression highlighting times when 'drought' received much mention (and thus attention) through written sources, such as during the significant drought events of 1865-69, 1877-79, 1895-96 and 1900-03. Although the 1900-1903 event does not receive as much mention (according to Figure 2) as those during 1895-96 and 1877-79, this is largely due to fewer documentary source materials having been consulted for times since ~1900. The more recent documents contain a much greater detail of information, hence requiring fewer sources. However, the figure also demonstrates that concerns of perceived drought conditions are reported much more frequently (66% of years) than the actual occurrence of drought (29% of years) during the 19th C. This is largely due to conditions of [prolonged] seasonal aridity, usually described as 'drought'. Table 1 lists the reported consequences, concomitant

phenomena and human responses during each of the identified drought periods. We also
provide a brief overview on the spatial extent of these droughts through other parts of southern
Africa, using previously published 19[th] C documentary-based climate chronologies. Some
comparative emphasis is placed on the neighboring semi-arid regions of the Kalahari to the
southeast and east of central Namibia, and Namaqualand (winter rainfall zone) to the south of
the current study area (Figure 1).
One of the first droughts (1850-51) experienced by missionaries of the RMS resulted in
grasslands becoming degraded and barren, and eventually led to hunger, starvation and death
amongst the indigenous population (Hahn Diaries, 581). Missionaries were particularly
distressed that the majority of people left stations in search of food, and consequently, that very
few children attended school (ARRMS, 1850, 21). This drought was widespread across much
of southern Africa (Nash and Endfield, 2002), and was accompanied by famine and livestock
deaths in Lesotho and surrounding regions (Nash and Grab, 2010) (Figure 2). In Namaqualand,
drought conditions occurred in 1851 when the winter rains largely failed (Kelso and Vogel
310  2007).

The failure of two rain seasons (1858-60) carried consequences of widespread hunger, poor
harvests, livestock deaths and missionaries relying on food transported from the Cape colony.
Traveler and explorer James Chapman was in Otjimbingue on 1[st] January 1861 and comments:
"*No rain of any consequence has fallen here for 2 years. No grass anywhere, the trees and*
*bushes bare*" (Chapman, 1971, 217). Although this was a period of 'relatively dry' conditions
across central southern Africa accompanied by early and late seasonal rains but mid-summer
drought during the 1858-59 rain season (Nash and Endfield, 2008; Nash and Grab, 2010), it
seems that desiccation and its consequences were more pronounced over central Namibia than
elsewhere. To the south, in Namaqualand, conditions in 1859 were wet, but followed by
drought (1860-1862) for which the first known regional government assistance was proposed
(Kelso and Vogel, 2007). This demonstrates that periods of wet and dry are not always
synchronous between the mid- to late-summer rainfall region of central Namibia and the
predominantly winter rainfall region to the south (Namaqualand) (Figure 2).

The extended drought of 1865-69 ranks as the longest (four consecutive failed rain seasons)
over central Namibia between 1850 and 1920. On 7[th] February 1866, missionary Brincker
writes from Otjikango that: "*in this year there is a great drought as is seldom experienced in*
*this land, such that even the Swakop* [River] *has not yet* [7[th] Feb 1866] *come down* [or reached
Okhandja], *which otherwise would flow in December at the latest*" (VEM RMG 2.585 C/i 6,
63). Later it emerged that the Swakop River never reached Otjimbingue for three years (1866-
1868) (Irle, 1906, 22). What made this drought so devastating is the cumulative year-on-year
effect that progressively worsened the situation, leading to widespread hunger, starvation and
death of indigenous people. In the Kalahari, this period started as relatively dry but for the most
part was near normal (Nash and Endfield, 2008). However, winter rains largely failed in
Namaqualand for four consecutive years (1865-68) (Kelso and Vogel, 2007), indicating
prolonged drought over the westerly sector of southern Africa. Reports for central and eastern
regions of southern Africa were variable, with near normal to relatively dry conditions over
most parts, but some regions experienced harvest failures (Nash and Grab, 2010; Nash et al.,
2016). Noteworthy is that while there was widespread and prolonged southern African drought
over the summer rainfall regions between 1861 and 1863 (Nash and Endfield, 2008; Nash and
Grab, 2010; Nash et al., 2017), this period was relatively wet (1861-62) to very wet (1862-63)
over central Namibia (Figure 2). Then, when drought commenced over central Namibia during
the late 1860s, hydro-climatic conditions improved over most of the southern African summer
rainfall regions.

The 1877-79 drought affected most southern African summer rainfall regions (Nash et al.,
2019) and coincided with what has been described as the 1877-78 'Global Drought' and
'Global Famine' caused by a major El Niño (Davis, 2001; Hao et al., 2010; Singh *et al.,* 2018).
This was indeed one of the most devastating droughts in recorded history over central Namibia.
This drought, in connection with increasing conflicts that had complex causes, had multiple
consequences (Table 1): crop failures, obliterated grasslands, dead trees, lack of wild foods,
social tensions and stock thefts, collapse of commercial enterprises, poverty, starvation and
death amongst people and their livestock. Missionary responses to this drought included
dedicated days of prayer and repentance, and fundraising so that food could be purchased for
those in most desperate need. By 1879 the "*conditions in Hereroland* [had] *not improved, but*
*in the contrary, the longer the worse it* [had] *become. By far the main cause of this* [was] *the*
*endless drought* [….] *it seems that every now and again such periods return to southern Africa,*
*where the drought worsens with each year, as is the case with Hereroland now, which finds*
*itself at the end of a whole number of such years.....*" (ARMS, 1879, 19f). This drought seemed
even more prolonged (1877-81) in the Kalahari but was not spatially synchronous across this
region, with one or more isolated reports of good rains in early 1880 (Nash and Endfield, 2002).
Drought conditions prevailed over central and eastern southern Africa during the years 1876-
79, with reports of poor crop production over Lesotho (Nash and Grab, 2010; Nash et al., 2016).
However, in direct contrast to the summer rainfall regions, 1878 was a wet year over
Namaqualand (but again dry in 1879).

The situation associated with the 1877-79 drought, in most places of central Namibia repeated
itself in 1881/2, largely owing to the combined effects of drought and war (for a more detailed
description see Grab and Zumthurm, 2018). The drought of 1887-90 was again a lengthy one
with similar consequences to those previously. Only the poorest of people stayed at mission
stations, who resorted to begging for food. Others had again spread out and followed a nomadic
lifestyle in search for grazing and water. Large stock losses were reported from mission
stations, while much of the indigenous population remained in a state of poverty and hunger
(Table 1). This drought was one of the least synchronous across southern Africa during the
latter half of the 19th C. The Kalahari was relatively wet to relatively dry (Nash and Endfield,
2008) and Namaqualand normal to wet (Kelso and Vogel, 2007). Although easternmost
southern Africa experienced one of its most prolonged droughts of the 19th C (1886-90) (Nash
et al., 2016), further inland (Lesotho and central South Africa), conditions ranged from
relatively wet to relatively dry (Nash and Grab, 2010). In the extreme northern parts of southern
Africa (Malawi), conditions during this time were initially relatively wet (1885-87) but drought
commenced during 1887-88 (Nash et al., 2018).

The final drought of the 19th century to impact central Namibia was due to the failed 1895/96
rain season. Rainfall records indicate only 48-50% of normal seasonal rains falling over the
central and northern regions, while to the south at Rehoboth only 44% of the norm was
measured (Figure 2). According to the Annual Report of the RMS, "*in the entire Southwest
Africa there* [was] *a major drought over most of the year, and in the southern parts of the
country, the so-called Gross-Namalande, it caused total famine.* [They] *thus had to raise funds
*[….] *to avoid starvation*" (ARRMS, 1896, 14f). Cattle and draught oxen were reportedly in a
very weak state, and to make matters worse, the Rinderpest (cattle plague) had arrived which
further decimated stock. In this case, the drought was synchronous across southern Africa and
considered one of the most prolonged (1894-99) and severe during the 19th C in the Kalahari
(Nash et al., 2016). Relatively dry conditions prevailed over central southern Africa (Nash and
Grab, 2010), but along eastern South Africa drought prevailed (1895-1900) with severe food
shortages due to poor crop yields, excacerbated by locust infestations and the Rinderpest (Nash
et al., 2016). This led to a variety of socio-economic consequences across broad regions of

eastern and central southern Africa, such as poverty, malnutrition, migration and socio-ecological change (Pribyl et al., 2019). This also coincided with the longest period of consecutive dry/drought years in Namaqualand (1890-99) during the 19[th] C (Kelso and Vogel, 2007). Although dry conditions prevailed as far north as Malawi until 1894, wetter conditions returned to that region thereafter (Nash et al., 2018).

The period 1900-03 was characterized by three successive below-average rainfall seasons (averaging ~62%, 55% and 60% of the norm respectively for central Namibia) (Figure 3). The impacts were again cumulative with each year, in particular affecting groundwater and grazing. What made this drought worse still, was the ongoing Rinderpest (despite vaccines now being used), outbreak of Texasfever among cattle, and repeated locust invasions which decimated any new grass growth and crops after it had rained a little. The Otjimbingue 1901 station chronicle summarizes the situation after the first of these failed rain seasons: "*The drought lasted until early March* [although it continued to be dry thereafter]. *The people's gardens were desiccated without exception, hunger was great, especially given that no employment was possible at this place. The wells are drying up and the spring for the mission houses has had no water for many weeks* […] *In February we had three rain showers which totalled 59mm. The river came down very weakly for two days, enough to provide some water to the wells. Consequently, it started to green up in the area. But alas, the blazing sun and locusts soon destroyed the greenery. The follow-up rains never came and so the long period of drought continued*" (VEM RMG 2.588 C/i 8: 355f.). The extended drought became so bad that it resulted in some mission stations having to close down (something not reported during previous droughts), such as the one at Omandumba (ARRMS, 1903). This was a widespread southern African drought, with reported crop failures (Thorp, 1926; Msangi, 2004; Manatsa et al., 2008).

According to the 1907/08 Annual Report for Southwest Africa, "*The rainfalls were not very productive. In April and May 1907 there were abundant rainfalls so that the grazing and water situation was good. In contrast, rainfall in this last season was well below average. Even though this had less consequence on grazing to the north, the water situation was unfavourable, so that on many farms there were complaints about lack of water even at the beginning of the dry season*" (NAN, ZBU, 155 A.VI.A.3, vol. 17, 232). Overall, central Namibia only had on average ~69% of its mean rainfall. Some places received near-normal rainfall, and thus did not suffer drought (e.g. Otjimbingue received 88% of its normal rainfall). Other areas, however, experienced drought conditions, such as Rehoboth (which received only 58% of its normal

rainfall) and Omaruru (where the river never flowed during the rain season and the water
situation was dire) (ELCIN, V.23.1, 351). In contrast, there were reports of good agricultural
outputs over other parts of southern Africa with no mention of drought (Thorp, 1926).
However, for the Karoo region of South Africa, the year 1907 was identified as the start of a
near continuous run of below average rainfall, which lasted until 1923 (du Toit and O'Connor,
436 2014).

The drought of 1910/11 was particularly severe given far below normal rainfall during the rain
season, affecting all regions of central Namibia. According to the Annual Report for Gobabis,
*"The rainfall season of 1910/11 was very bad. Especially for farming, as the December-*
*January rains were almost entirely absent – only in March was there abundant rain* (Annual
Report for Gobabis, 1910/11, 42f). The instrumental records support this, indicating only 10%
(Otjimbingue) to 26% (Gobabis) of normal Dec/Jan rainfalls across stations. Although some
late season (March-May) rains indeed fell at Gobabis (100% of the norm), all other stations
recorded well below normal late season rains (17% at Otjikango to 44% in Windhoek). This
drought carried severe consequences, such as large stock losses (also due to the Rinderpest),
near complete harvest failures, and a desperate shortage of water for human and livestock
needs. Drought was also reported from South Africa (1909-11)(Msangi, 2004), while the year
1911 marked the start of a long dry spell (1911-1916) in former Southern Rhodesia
(Zimbabwe) (Manatsa et al., 2008).
The drought of 1912-13 was again widespread, as also confirmed by the instrumental rain
records (Figure 3). Since rainfall records began in 1891, this was the driest rainfall season in
the south (Rehoboth: 33% of the norm), 3$^{rd}$ driest in the central highlands (Windhoek: 66% of
the norm) and 2$^{nd}$ driest in the north (Okahandja: 45% of the norm), and this collectively must
rank as one of the most severe droughts (in terms of rainfall/water deficit) since the mid- 19$^{th}$
century.  Such conditions are confirmed in the Otjimbingue station chronicle for 1913, which
describes the land "*far and wide looking dreary and burnt* [by the sun]", but that the mountain
areas had received some rain (VEM RMG 2.588 C/i 8, 415).  The grazing situation was critical
at Otjimbingue, with apparently "*not a single halm of grass to be seen for many hours distance*
*from the station*" (ARRMS, 1913, 40f), and around Rehoboth in the south where "*even the*
*hunter gatherer communities could not find the essentials to keep themselves alive*" (ARMS,
1913, 14.). The drought was characterized by complete crop failure in some areas and meagre
crop harvests in others, widespread drying up of wells, and depleted grazing, such that farmers
were preparing to vacate their land. This drought was synchronous over most of southern Africa
(Thorp, 1926; Manatsa et al., 2008; du Toit and O'Connor, 2014).

**4.2 Sub-regional rainfall variability**
Strong rainfall gradients occur through central Namibia, both north-south and west-east (Figure
1), which, together with 'patchy' (isolated) rainfall distribution in some years, may at times
account for strongly contrasting sub-regional conditions (Figures 3 & 4). Thus, while most
drought events affected the entire region, there were several instances when one or more areas
had 'sufficient' or 'relatively wet' conditions during a 'regional drought'. One or two isolated
heavy rain showers in a particular area may have been enough to permit local stream discharge
and rapid grass recovery, while surrounding areas remained parched and dry. For instance, the
rain season failed entirely in Otjimbingue in early 1868 and grazing conditions were in a
terrible state, yet some rains fell and streams flowed three times in Omaruru further north,
where there was sufficient grazing, vegetable gardens could be set, and corn be planted (BRM,
1868, 355). Missionary Heidmann reports from Rehoboth on 27 December 1877 that they had
not suffered as much from the drought as those at other stations across central Namibia. Given
that the drought impact at this usually drier locality was not as severe as that at usually better
watered regions, may imply that Rehoboth had rainfall closer to its norm than at other regions
(VEM RMG 2.589 C/i 9, 143). The 1895/6 rain season over most of central Namibia was dry,
but further south (Rehoboth southwards) became critically dry with drought conditions. Yet,
the usually much drier western region of Otjimbingwe had abundant rain, so much so that
"*grass over the new year was so good, as was not seen in many years*" (ZBU, 146, A.VI.A.3,
vol.2). During the drought of 1900-03, conditions were at first also reported to be variable
across sub-regions. For instance, towards the end of 1901, while the much awaited rains had
arrived in the northern regions, these were apparently scanty/patchy in the southern parts
(ARRMS, 1902, 24). However, while the end of year (Nov/Dec) instrumental rain records for
1901 indeed show high rainfall in the north (Okahandja: 156% of the norm), they also show
slightly above normal rainfall for central (Windhoek: 110% of norm) and southern (Rehoboth:
115% of norm) station localities. At other times the documented accounts compare positively
with the instrumental records, such as was the case in 1910, when apparently abundant rains
fell at Omaruru (northern study region), "*but in other regions of the land it was not favourable
in this regard*" (ELCIN, V.23.1, 375). Instrumental records confirm this, with Okahandja
receiving 110% of the normal rainfall, while western, central and southern regions (Otjikango,
Windhoek, Rehoboth) only received between 75-80% of normal rainfall. However, Gobabis in
the eastern part of central Namibia received 122% of its normal rainfall in 1910. This
demonstrates that in addition to the strong rainfall gradients across the region, there were also
disparate rainfall departures from the mean in a given season or year. In this case, the somewhat
wetter regions to the north and east received above normal rainfall, while the drier regions to
the west and south received less than normal rain, consequently exaggerating rainfall gradients
even more beyond their norm.

Conversely, there were times when most of central Namibia experienced 'relatively dry' to
'near normal' conditions that would not qualify as a drought. In such years, most areas received
sufficient rains but there were instances when sub-regions experienced drought. The year 1890
started variably; in Otjimbingue, 100km south of Omaruru, the rains failed, causing people to
disperse (RMG 2.588 C/i 8, 307), yet at Omaruru, sufficient rain had fallen to permit good
grazing conditions, such that people congregated at the station again (ELKIN, V.23.1, 160).  In
early 1891, Otjimbingue and Okombahe again had drought while reports from other regions
confirmed that good rains had fallen (RMG 2.588 C/i 8, 312).

**5.  Discussion**

What follows is a discussion on how missionaries perceived and experienced droughts and
their consequences through the time-period 1850-1920. Sub-periods of time are unpacked and
characterized according to the most notable and written about impacts. This does not suggest a
rigid linear development of drought impacts and responses through time, and neither do we
imply that one particular impact was restricted to a given sub-period. Rather, the intention is to
demonstrate that the impacts, consequences, responses and perceptions of drought during this
historical period were not static through time.

**5.1 Drought during the 1850s: from famine to societal dispersal**

Missionary Hahn, stationed at Otjikango, reports the first drought-induced famine during
spring 1851. First reports of deaths from starvation date from September 1851, and on 19
October Hahn wrote in his diary that the "*misery is enormous. Almost daily you see new pitiful*
*creatures arrive at the station. They drag themselves over here to get some food. Our help is*
*not enough at all*" (Hahn Diaries, 515.). On 9 November 1851, Hahn noted that several children
had died and that the hardships were severe owing to terrible drought.  By mid-December he
observed that there were more victims of drought and hunger and that not even a third of the
missionary station inhabitants remained, but that people had scattered into the 'veld' (open
country) where they were in search of wild berries and roots. It was only towards the end of
December 1851 when rains finally arrived, but these were too late to avoid further hunger and
starvation. From Rehoboth, missionary Kleinschmidt expressed concern at the absence of
many children from school due to drought and the dispersal of people. During 1850, some 180
pupils attended classes, but dwindled to only 70 learners by April 1851 (ARRMS, 1851, 23).
On 22 June 1852, missionary Rath wrote from Otjimbingwe that "*the people who remain are*
*parched by hunger and stray around like hungry wolves. You cannot do anything with such*
*people anywhere in the world, least of all among pagans. The needs of the stomach overshadow*
*everything else*" (VEM RMG 2.588 C/i 8, 36).

The tension for missionaries during this time was that while their calling was to attract people
to the stations for evangelistic and educational purposes, they did not have the capacity to feed
local inhabitants during times of drought and crop failure. Hence, people resorting to hunting
and gathering during such times, which meant dispersal of the population, and mission stations
being deserted.  Similar tensions are aluded to by Endfield and Nash (2002) for the Kalahari
region, where the nomadic lifestyles of indigenous people during earlier decades of the 19[th] C
meant finding strategies to attract local populations to permanent settlements. In central
Namibia, the missionaries themselves were in dire need of food and lacked any institutional
supporting structure to assist them during times of severe food shortages. For instance, when
missionary Hahn travelled past Rehoboth station on his way to Cape Town in 1859, he was
shocked that missionary Kleinschmidt and his family could only drink goats' milk and
depended on food they received from travellers. Their cattle were too malnourished to provide
milk or meat (ARRMS, 1859, 34).

Population dispersal and movement as a local drought/famine coping mechanism would not
have been a new thing and was a typical/logical response that would continue into later decades
(Table 1).  During times of drought, dispersal (transhumance) was generally towards the better
watered north and northwest, but was likely restricted in distance given that such regions would
themselves already have been inhabited. A similar, but more regular form of transhumance was
observed during the first half of the 19[th] C among the Namaqua Khoikhoi people of
Namaqualand (Kelso and Vogel, 2015). Migration between the winter rainfall regions of
Namaqualand and the summer rainfall area of neighboring Bushmanland served as a form of
resilience and coping mechanism to overcome the impacts of drought in that region (Kelso and
Vogel, 2015). Although such human movement in response to 19th C droughts is less widely
reported from the wetter regions of the sub-continent, it is reported that the combined impacts
of drought and Rinderpest in the mid 1890s, resulted in the abandonment of villages and large
scale migration in some of these regions (Pribyl et al., 2019).

**5.2 Drought during the 1860s: from dispersal to societal tension**

Drought during the 1860s intensified and that of 1865-1869 was one of the longest and most
devastating during recent historical times (Grab and Zumthurm, 2018). During this 'great
drought', missionary stations were again vacated, as even missionaries and colonists
themselves were forced to abandon the stations. For instance, economist Redecker departed
Otjimbingue with some of the converts to relocate where surface or ground water was still
available along the Omaruru River (VEM RMG 2.588 C/i 8, 199). Others that remained at
their station (e.g. missionary Viehe, see below) felt that they had been abandoned and left in
need by the absence of all those who had left. Brincker reported from Otjikango on 10
September 1869 that "*the drought and in its wake the famine is pushing very hard on us and*
*many poor people have died of starvation. Indeed, it was told here, that the hunger among the*
*Ovatjimba or the poor Herero is so large that they resorted to cannibalism, which most likely*
*is exaggerated*" (VEM RMG 2.588 C/i 8, 70). This is the only account which hints of
cannibalism in all the documents analysed, the reality of which even the missionaries doubted.
It thus serves to emphasise the seriousness with which the situation was viewed. In desperation,
missionary Brincker also departed Otjikango station and moved to Otjimbingwe where
missionary Hahn was stationed. Here too, there were only a few men with their families who
remained. Despite the shortage of food, Hahn claims that he was left with little choice but to
feed some hundred children from money provided by the missionary society (BRM 1869,
262f). While there had been some improved institutional financial support from Germany by
the late 1860s, such support seemed insufficient to benefit the needs of those residing at
stations.

Missionaries usually demonstrated sympathy towards their communities and the nomadic
habits of their people. Although missionaries expressed a deep understanding of the tensions

and needs faced by the local population, their descriptions began to portray an undertone of disdain towards what was considered 'unChristian-like' behaviour. For instance, in May 1868, missionary Viehe complained from Otjimbingwe that most of the residents were away and would thus not be able to care for him and his family, and writes: "*but who can take this amiss for a pagan people?*"(BRM, 1868, 247).  Drought seemed to regularly interrupt the core purposes of the RMS in central Namibia, as is reflected by missionary Brincker from Otjikango towards the end of the long drought (August 1872):

"*There is one thing that worries me, although an earthly one, it is the drought that is increasing each year. What should become of our communities if they cannot settle down and hence consolidate? Admittedly, we cannot complain about the roving of our community members, but the question arises if it is possible at all to implement culture under such unfavourable circumstances. The nature of this country treats these poor people more than uncharitably*" (BRM, 1882, 234f).

Drought during the late 1860s was accompanied by armed conflicts, which seemed to have escalated with time. Hence, human movement to and from mission stations was no longer only a consequence of drought but also due to conflict.  Missionaries were well aware of this, so that in the annual report of 1869, war was identified as the primary reason for the scattering of residents from Otjimbingwe. The editor added: "*we hope for peace and rain so that the bulk of the blacks can move onto the station again and our missionaries are saved and full of work again*" (ARRMS, 1869, 24).  Missionary Heidmann, who had just re-opened the station at Rehoboth in 1871, acknowledged that it was not only the long drought and associated general scattering of people, but also the "*endless clan feuds and plundering raids*" that were responsible for the impoverishment of the once wealthy community (BRM, 1871, 129).

Drought and conflict cannot be separated in such circumstances as it was the scarcity of grazing resources, death of livestock, hunger and starvation due to drought, that essentially lead to many of the conflicts, wars and livestock thefts. These were also connected to increasing trading activities and wealth accumulation in the form of cattle (Henrichsen, 2011; Wallace, 2011). In Namaqualand, local communities experienced an aggravation of their material situation at the same time, even though conflicts of the same scale did not occur there. However, people lost much of their cattle and land to new settlers (Kelso and Vogel 2015). This development decreased their mobility and increased their dependence on agricultural output, consequently reducing their ability to deal with climatic stress. In central Namibia,

mobility remained a crucial strategy to overcome drought, despite complicated interactions
manifested through political and armed conflicts.


**5.3 Drought during the 1870s: from societal tension to environmental deterioration**
The effects of armed conflicts became even more pronounced during the drought of the late
1870s, a particularly severe dry period which affected most of southern Africa (see Nash et al.,
2019). To make matters worse for the missionary vision was that the exodus from stations
continued during periods of drought. The year 1877 was not an easy one for central Namibia
(known as Hereroland at this time): "*firstly there was a long drought with famine",* and
secondly because of *"a strained relationship between the Herero* [indigenous people group]
*and British colonists"*. In addition, the Namaqua [another indigenous people group] had to deal
with their loss of power. Collectively, these factors triggered conflict, which, *"together with*
*the consequences of drought increased distress and want even more"* (ARRMS, 1877, 19f).

In 1877, William Coates Palgrave was sent as a special commissioner from the Cape to
investigate whether Namibia had potential to become a valuable British colony. He commented
on the extensive drought after arriving at Walvis Bay on 12[th] October 1877: "*The drought*
*which has so seriously affected the Colony has also been severley felt in this country and Great*
*Namaqualand, particularly by those who are wholly or in part dependant on the wild products*
*of the earth for their subsistence. Many of those are starving and stock-lifting has become*
*unusually prevalent and has given use to much bad feeling between the tribes"* (CAD, NA 286).
Many contemporary observers noted that the Herero's cattle had rapidly multiplied over the
years. They moved southwards in search of new pastures due to drought in northern Namibia,
although political motives also played a role (Henrichsen, 2011). Missionary Heider from the
southernmost station of the study area, Hoachanas, wrote in 1877 that the complete Nama
community was forced to leave the station due to the Herero pushing into the region with large
herds of cattle (ARRMS, 1877, 31). Missionary Büttner, who had spent seven years at
Otjimbingwe, predicted in the same year that the expansion of the Herero would force the
Nama and Damara to become "*violent thieves*" (BRM, 1878, 11). A year later (1878), it was
estimated that some £800 worth of stock had been stolen over a 6-month period in the
immediate surrounds of Rehoboth (VEM RMG 2.588 C/i 8. 247).

Due to a seemingly endless drought and armed conflict, conditions in Hereroland progressively
worsened through the period 1877-79. The impression was that due to multiple drought years,
conditions had worsened with each year in an accumulative manner, such that inhabitants
suffered greatly. So much so, that this led to much conflict between white settlers and the
indigenous Herero over want for the little grazing still available. Conflicts also arose between
the Herero and Namaqua, as also between English border patrols and those moving their herds
(ARRMS, 1879). At this stage, and continuing into the early 1880s, the entire German
missionary cause in central Namibia seemed to have disintegrated and required new approaches
given the constant coming and going of local people, in response to war and drought.
Missionary Brincker writes from Otjimbingue (1882): "*There are two extremely obstructive*
*enemies to our work here, namely war and drought.* [….] *Our people have received a wretched*
*land for their inheritance, in which no culture is possible. Christianity must take on a new form,*
*it must nomadize, which has probably not yet been sufficiently understood and considered*"
(BRM, 1882). Missionaries at various stations responded with a declaration to commit one
hour of prayer for rain, twice monthly.

The impression from missionaries was that drought had so much reduced wild foods (bulbs,
roots, berries, game and "creeping things") that the Damara (mostly hunter-gatherer
communities) were forced to steal livestock to stay alive. Missionary Bernsmann from
Otjimbingwe, for example, wrote in 1878 that the Herero cast out the Nama and the Damara
from their places and that "*there was only very little food to gather in the fields and* [that] *the*
*game* [had] *escaped to places out of reach where they would still find good pastures. What*
*choice other than stealing do they have?*" (VEM RMG 2.588 C/i 8, 247). This led to campaigns
between the Damara and Herero, with "bloody consequences". The views of the German
missionaries was, however, that the situation would not have been as bad had it not been for
the English governments' plans to colonize Hereroland (ARRMS, 1879, 19f). They were,
nevertheless, also very critical of the indigenous population for what was perceived to be
overstocking. On 13th March 1879 missionary Büttner makes a written complaint to the local
inhabitants near Otjikango: "*….in earlier times when you had less livestock you could stay at*
*one place, and I remember in times of past drought how the church and school was full. Now*
*that you are wealthy* [with livestock] *you always complain of hunger and avoid coming to the*
*station*" (BRM, 1879, 302).

Notably, German missionaries gave the Damara considerably more attention during the
drought of the late 1870s than during that of the preceding decade. Several missionaries
emphasised the particularly hard fate of these people. Due to the failure of rains and more
intensive hunting of wild animals and gathering of edible plants, it was the widespread
impression that such *wild food* products became increasingly scarce. Similar observations (i.e.
disappearance of wild foods after drought events) were reported from the Kuruman region of
the Kalahari during the 1850s, where the environment and settlement history is similar to that
of central Namibia (Jacobs, 2002). By the 1890s, environmental deterioration (e.g. dearth of
wild edible plants and animals) seemed widespread across southern Africa and consequently
impacted drought-resilience amongst indigenous communities (Pribyl et al., 2019).

Endfield and Nash (2002) discuss in some depth the considerable attention given by
missionaries, such as David Livingstone, to desciccation theory. Missionaries and travellers
attempted to explain the reasons for what they viewed as progressive dessication of the
Kalahrai region. Although similar concerns were at times expressed by missionaries in central
Namibia, these were usually in response to a particular extended period of drought. More
notable, however, were concerns for environmental deterioration – which itself was strongly
linked to depleting water resources. Rapid environmental deterioration during the 1870s not
only constituted the depletion of wild edible plants and fauna, but also groundwater resources.
Missionaries, colonists and indigenous peoples relied heavily on perennial springs, and
particularly so through the long dry seasons. Although unsustainable water extraction and
harvesting of wild foods is already alluded to in the 1860s, such accounts become much more
prominent during the 1870s and subsequent decades of colonialism. On 11[th] October 1860,
missionary Rath arrives at Tsaobis station and comments that this place formerly had a spring
that never dried up. He laments that the nonsensical economy of the whites resulted in "*not a*
*drop of water to be found there anymore*" (VEM RMG 2.588 C/i 8, 117). A decade later
(September 1871), missionary Hahn writes from Ameib, reflecting that in past years, water in
abundance had occurred there and in the Erongo Mountains, but that given the severe droughts
over the past years, there had been dramatic disappearance of springs. However, he also blames
the Namaqua people for the general environmental destruction, particularly the deforestation
of shade bearing mimosas (VEM RMG 1.577 a B/c II 3, 451). By late February 1877,
missionary Dannert at Otjimbingwe noted that the spring, which usually had running water
throughout the year, had dried up. Water was only available at a depth of seven feet. Earlier
there were rows of poplars growing in front of the mission house at Otjimbingue, but these, as
most of the fruit trees planted by missionary Hörnemann, had perished by 1877 owing to
drought (RMG 2.588 C/i 8, 242f). Otjimbingue, Omaruru, Omburo, and other mission stations
had 'permanent' springs in their riverbeds, from where water flowed onwards for at least an
hour's walk during the entire year. However, by 1879, such spring water had dried up
considerably, or even disappeared in some cases. Consequently, one now had to dig wells in
the Otjimbingue and Omaruru streambeds, while the spring at Omburo only flowed over half
its former distance (ELCIN, V.23.1, 63).

**5.4 Drought during the colonial era (1880s-1920): capitalism and further**
**environmental deterioration**
Gradually, during the 1870s, opportunities for wage labour expanded more rapidly. One of the
first mentions of wage labor comes from missionary Böhm stationed at Ameib in 1873:
"*Hunger and poverty belong to the lives of the Namaqua, but one can sense that the desperation*
*is no longer as severe as in previous years. Most of these people, apart from during short*
*hunting campaigns, tend to stay at the station even during dry times. The men earn much*
*through ostrich hunting and last year made plentiful tobacco, a portion of which they sell*"
(ARRMS 1873, 37). The increasing dependence on wages had positive and negative
consequences for the ability of indigenous inhabitants to acquire food. It diversified their
livelihood options and, as also reported from eastern parts of southern Africa (c.f. Pribyl et al.,
2019), alleviated stress on local food supplies. In contrast, during earlier 19$^{th}$ C drought events
in central Namibia, missionary stations were the primary (and often only) source of food aid to
those most in need. However, this diversification did not noticeably increase their resilience to
drought. In part, this is because they became more vulnerable to harvest failures as community
and family structures were weakened (c.f. Pribyl et al., 2019) and less time was invested in
subsistence agriculture. Similar consequences of externally exposed and novel economic
realities were observed in late 19$^{th}$ C Namaqualand (Kelso and Vogel, 2015).

One of the most important new modes of earning a living for people connected to missions was
the so-called *Frachtfahren*, which involved the transporting of goods by ox-wagon (ELCIN,
V.23.1, 51). However, *Frachtfahren* was interrupted in 1878 due to drought (lack of water and
food for draught oxen) – this had serious implications for those reliant on wage labor. As
commerce increased, many new drivers were required by the 1890s. The head of the
Otjimbingwe district reported in 1897, that while indigenous people had extensively cultivated
crops in riverbeds in earlier years, this practice had receded in importance given that
considerable money could be earned through *Frachtfahren.* Consequently, it was more
attractive for drivers to earn a living and buy food, rather than to produce it themselves (NAN,
ZBU, 147, A.VI.A.3, vol.2a., 142). This practice was not without its problems, especially after
the Rinderpest. People had lost their livestock during the outbreak and were now forced to buy
goods or new oxen on credit. A similar situation troubled communities further south in
Namaqualand during the 1860s (Kelso and Vogel 2015).

During the 1900-1903 drought, there were several accounts of people not having enough food
in Rehoboth, Omaruru and Otjimbingwe given the fact that income opportunities from
*Frachtfahren* had declined, also due to drought (ELCIN, V.23.1, 245; ARRMS, 1901, 24;
VEM, RMG 2.588 C/i 8, 355f). For 10-11 months the drought was so severe that the
*Frachtfahren* closed down almost entirely, and where it continued, it was at 'great loss'
(assumably loss of draught animals) (ARRMS, 1903). At the time, it proved difficult to find an
alternative way to obtain food. Prices were exceptionally high in times of drought, wild foods
were now increasingly scarce to find, and wage labourers generally did not cultivate crops
themselves. One possibility for supplementary wages during times of drought was to work on
the railways or in the mines for a meagre salary (ARRMS, 1911, 35; ELCIN, V.23.1, 252). In
Otjosazu, the harvests of 1901 largely failed, resulting in substantial hunger amongst poor
people who, unlike the more financially privileged, were unable to purchase food to replace
what they had lost through the bad harvest (ARRMS, 1901, 29).

A new form of relief for mission communities during the 1900-1903 drought was financial or
material support from the colonial government. The RMS mentions in its 1902 annual report
that the impact of drought was felt as severely as ever. The RMS thanked settlers and, in
particular, the German government for their support, through which stations had apparently
received not only drought relief money and food aid, but also financial assistance for much
needed infrastructural developments and renovations, which could improve future drought
coping mechanisms (ARRMS, 1902, 20). For example, the station of Hoachanas received food
worth 1000 Mark from the German state, which, in addition, financed the construction of 22
wells (ARRMS, 1902, 20). The first reported construction of a sand dam/water reservoir is
mentioned in the 1901/02 Annual Report for the Windhoek district (p228). Water in this
reservoir had apparently reached a depth of 3½ m in 1902 and demonstrates a first major
infrastructural and long-term water management initiative. It is doubtful, however, that such

government aid had any far-reaching positive effects as many people were still forced to find wild food products during times of desperation and the general decline of human health was widely reported during the first decade of the 20[th] century. The official German Annual Report for the colony of South-West Africa (1911/12) announced that "*the lack of fresh milk, on which locals have depended as staple food for generations, plus the scarceness of field crops, which were the only available fresh vegetables for locals after the drought of 1911, can be regarded as the main reason for the many cases of scurvy*" (NAN, ZBU, 161, A.VI.A.6, vol 1, 16f).

### 5.4.1 Impacts on vegetation cover

Degradation of vegetation during times of drought seems to have been spatially patchy, largely owing to anthropogenic factors. Grass and shrubs were heavily grazed around mission stations and settlements where some water was still available (through springs, wells), as also along the transport routes. There are thus accounts of livestock deaths along transport routes for lack of grazing, such as was the case during the drought of 1877-79. On his journey from Ameib to Walfish Bay in March 1878, missionary Böhm described that there was no grass to be seen along the route, and even less so at watering points and grazing posts. He observed oxen from many other people on their way to collect goods from the ship (at Walfish Bay), but that many of these had died as they were too starved and weak – many lost more than half their outspan (BRM, 1878, 206). As also mentioned by Grab and Zumthurm (2018), drought and war forced the Herero to keep their livestock close to Omaruru during the 1880-82 drought. Consequently, not only was grass cover completely depleted, but even grass roots were damaged due to trampling. This would have had longer-term consequences for vegetation recovery even when the rains returned. Once the situation had become more peaceful, livestock could be taken to more remote outposts where there was still sufficient grazing (ELCIN, V.23.1, 101). Similar accounts came from other stations during droughts and dry periods of the late 19[th] century, in part, also due to the substantial growth in livestock numbers. Missionary Diehl reports from Okahandja in September 1886 that grazing was so heavily depleted around the station that even soon after the end of the rainy season there was no grazing to be found in a wide area around the post (BRM, 1887, 75). Similar developments occurred in late 19[th] C Namaqualand, when, after decades of intensive land-use, it took communities much longer to recover from droughts than earlier in the century (Kelso and Vogel 2015).

Such situations described above would further worsen as livestock numbers continued to
increase and severe droughts return in later years. At the same time, trading intensified and
more and more goods were transported. On arrival of the 1895-96 drought, authorities had
realized that both the decimated vegetation and its associated risks to draught animals along
the northern transport route and its outposts via Otjimbingue, required some intervention (long-
term coping/adaptation mechanism). Thus, plans were made for an alternative more southerly
transport route, via Rehoboth:
"*With the start of the new year [1895] the heat intensified, and as a consequence also the*
*drought. Often the clouds accumulated and promised much rain, but the westwind blew them*
*away. The desperation increases, people and livestock suffer. The Frachtfahrer are afraid to*
*journey to the Bay because their losses increase from week to week [……] From Swakopmund*
*and the Bay, there have been some 880 freight items delivered into the hinterland in one year,*
*of which over 500 were transported via Otjimbingue. Some 10 000 to 12 000 oxen as draught*
*animals came over Otjimbingue this past year, where they would spend several days to rest,*
*feed and recover, but at the same time decimated the grazing. The troops have thus started*
*building an alternative rout via the Kuiseb River from the Bay to Rehoboth, and thereby relieve*
*the pressure on the main route from the coast to Windhoek*" (VEM RMG 2.588 C/I: 8).

**5.4.2  Impacts on groundwater**

Water management was an integral part of missionary life in southern Africa, particularly in
drylands such as the Kalahari, where wells and small-scale irrigation schemes were already
established in the 1820s (Endfield and Nash, 2002). Similar initiatives are documented for
central Namibia, but these were temporally considerably delayed in comparison to parts of the
Kalahari. Drought at the beginning of the 20th C had serious impact on groundwater availability
across central Namibia and wells drying up were widely reported, much more so than during
previous droughts (Table 1). For instance, the well at the missionary house at Otjimbingwe,
completely dried up in March 1901, preventing the planting of crops (VEM, RMG 2.588 C/i
8, 355f).  The missionary well at Omaruru, which "*always had water in abundance*", had to be
deepened in 1901 (ELCIN, V.23.1, 252).  The drought of 1901 was similar in magnitude (i.e.
rainfall quantity) to the drought of 1896 in most areas (Figure 3). This suggests that increasing
water demands and its associated groundwater extraction may have contributed to the faster
depletion of groundwater in 1901, and hence the necessity to go deeper. Accounts of
springs/wells drying up became frequent during the colonial period, even during 1903/04 when
rainfall had improved slightly in some districts (NAN, ZBU, 151, A.VI.A.3, vol.10, 102;
Annual Report 1903/04, Windhoek). After another dry rain season (1907/08), the head of
Windhoek district reported that numerous wells were dry (NAN, ZBU, 156 A.VI.A.3, vol. 19,
3). Although wells were deepened at Omaruru in 1907, the following year, missionary Dannert
had to dig even deeper to reach water required for domestic purposes. The situation worsened
during the drought of 1910/11, forcing the colonial government to increase drilling activities
and go deeper still. In early 1911, the great well at Otjimbingwe, which was by now operated
using a wind-engine, had dried up for the first time since its construction 35 years earlier. The
stations first Herero Christian convert, Johanna Gertse (75 years of age) could not remember
the water-level ever being that low (VEM RMG 2.588 C/i 8, 405). Such accounts further
suggest rapid groundwater depletion during the early 20$^{th}$ century due to recurring droughts
and greater water extraction driven by both water demand and improved ability to do so. In
response to the severe drought of 1910 and associated state of emergency on farms, the German
colonial government committed itself to drilling operations on private farms. However, given
such a low water table, drilling was required to much greater depths than during previous dry
periods, in some cases to depths of 40-50m (NAN, ZBU, 159, A.VI.A.3, vol. 24. 85f). Reports
in 1911 emerged from many districts that blasting and drilling operations were being
undertaken in desperation to reach groundwater. For instance, in Otjikaru, drilling was required
to 38m depth, but even so 'only' provided 250 litres per hour (ARRMS, 1911, 37). A
consequence of wells is enhanced grazing resource and wild food depletion in the vicinity of
such watering points. The congregation of people and their livestock around such scarce water
resources during dry seasons and times of drought, has led to ongoing associated landcover
degradation during more recent times in semi-arid regions of southern Africa (c.f. Campbell,
890 1986).
While technological advancements during the first decade of the 20$^{th}$ century permitted water
extraction from greater depths, and served as both an immediate drought coping and longer-
term drought adaptation mechanism, this surely had negative implications for future
groundwater resources, water supply and ecosystems. During the severe drought of 1910/11,
apparently "*hundreds of large and strong trees along the Omusena River perished for lack of*
*water*" (VEM RMG 2.588 C/i 8, 405f). During recent times, similar concerns have been
expressed for riparian vegetation along Namibia's ephemeral rivers, where water availability
is erratic and sensitive to water abstraction and the construction of dams in upper catchments
(Jacobson et al., 1995; Jacobson and Jacobson, 2013; Arnold et al., 2016). We thus pose the

question whether this early ecological disaster (possibly the first reported in central Namibia) was due only to the exceptional drought, or a combination of drought and deep-water extraction associated with increased water demand?

## 6. Conclusions

This study has highlighted historical drought events in semi-arid central Namibia between 1850 and 1920. Early instrumental rainfall records (1891-1913) used in this study aid to quantify the hydro-meteorological severity of some of the identified drought events. These further demonstrate the confined period of summer rainfall (Dec-April) and the natural annual cycle of several months of negligible rainfall, constituting aridity rather than drought. Such instrumental rainfall records are valuable to quantify drier/wetter years, and the extent, duration and severity of droughts. However, determining the *real* impact of historical hydro-meteorological droughts depends largely on available documentary sources which report on environmental and human consequences and associated responses. To this end, the central Namibia historical drought context, within the given temporal and spatial context of this study, presents some important key findings:

1. The severity of historical drought impacts over central Namibia, during some drought events, were spatially strongly contrasting. This is given the extreme west-east and north-south rainfall gradients; hence percentage rainfall departures from the norm can be highly variable across the region during a given drought event. Consequently, place-based natural environmental and anthropogenic consequences and responses would differ markedly in magnitude during some drought events, as would reporting on the event.

2. Consequences of drought in a semi-arid environment with strongly seasonal rainfall are potentially far more catastrophic than drought events in regions with rainfall distributed throughout much of the year. This is due to the cumulative impact that a failed rain season has upon the subsequent long (~ 6 month) dry season. Our study also identifies multiple consecutive failed rain seasons (e.g. 1865-1869) that not only led to uninterrupted drought over several years, but also a year-on-year cumulative drought impact.

3. Human experience and associated reporting of drought events depends strongly on social, environmental, spatial and societal developmental situations and perspectives.

For instance, drought in this study is reported mostly from missionaries who were strategically positioned within the broader landscape (i.e. next to springs, episodically flowing rivers). Missionaries were relatively immobile given their career and societal calling. This would have been in direct contrast with the indigenous people groups, who led a highly mobile lifestyle across the entire region and beyond – although such mobility decreased through time and had dire consequences in later years (social tensions, conflicts, lowered coping mechanism to drought). As populations and livestock numbers grew, these resulted in overstocking (and overgrazing, excessive trampling) in specific spatial contexts with low carrying capacity during later years. Hence, the perceived impacts of droughts in later years would have also been a product of human engineered circumstances. In later years, increased water abstraction (lowering water tables), holding back river flow through reservoir constructions, the ability to more easily acquire imported foods, opportunities for employment and improved travel, would have collectively changed the dynamics and experiences of a given drought event. In addition, 'external' factors that were rare or unknown in earlier decades of the study period, but which became more prominent in later years (e.g. locusts plagues, Rinderpest, increased occurrence of fires) impacted human and livestock resilience, and thus perceived impacts of drought. This was not only the case over central Namibia, but also wetter regions of southern Africa (c.f. Hannaford et al., 2014; Pribyl et al., 2019). To this end, it is imperative to evaluate historical drought events, not only according to meteorological parameters, but also in consideration of changing natural-environmental and human-environmental contexts through time. For this, written-documentary sources are an essential and invaluable proxy record.

**Acknowledgements**

We thank two anonymous referees who provided valuable suggestions to help improve the manuscript.

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

Table 1: Reported consequences, concomitant phenomena and human responses to droughts
between 1850 and 1920 over central Namibia.

| | Droughts | | | | | | | | | | |
|---|---|---|---|---|---|---|---|---|---|---|---|
| Reported consequences | 1850-1851 | 1858-1860 | 1865-1869 | 1877-1879 | 1881-1882 | 1887-1890 | 1895-1896 | 1900-1903 | 1907-1908 | 1910-1911 | 1912-1913 |
| Hunger | * | * | * | * | * | * | * | * | * | | * |
| Starvation/human deaths | * | | * | * | | * | * | | | * | |
| Barren wasteland | | * | | * | * | * | | | | | * |
| Grasslands degraded / no grass | * | * | * | * | * | * | | * | * | | * |
| Trees/bushes bare | | * | | * | | | | | | | |
| Trees died | | | | * | | | | | | * | |
| Crop failures/no crop yields | | * | | * | | | | * | * | * | * |
| Lack of wild foods | | | | * | * | * | | * | | * | |
| Livestock deaths[1] | | * | * | * | * | * | * | * | | * | * |
| Wells dried up | | * | | * | | | * | * | * | * | * |
| Springs stopped flowing | | * | * | * | | | | * | * | | |
| **Concomitant phenomena and human responses** | | | | | | | | | | | |
| Population dispersal (vacated mission stations)[2] | * | * | * | * | * | * | * | * | * | * | * |
| Low school attendance[3] | * | | | * | | * | | * | * | | |
| Livestock thefts & social tensions[4] | | | * | * | * | * | * | | | * | |
| Farms vacated | | | | | | | | * | | | * |
| Closure of mission stations | | | | | | | | * | | | |
| Begging for food at stations | | | | * | | * | | * | | | |
| Prayers for rain | | | | * | * | | * | | | * | |
| Indigenous rain making[5] | | | * | | | | | | | | |
| Food aid from the Cape | | * | | | | | | | | | |
| Fund raising for food aid | | | | * | | | | * | | | |
| Colonial/governmental support | | | | | | | * | | * | | |
| Collapse of transport system | | | | * | | | * | | * | * | |
| Search for deeper wells | | | * | * | | | | | | | |
| Digging/construction of deeper wells | | | | * | | | | * | * | * | * |
| Construction of water reservoirs | | | | | | | | * | | * | |

Notes
1. Livestock deaths during droughts between 1895 and 1913 are due to the combined impacts of the cattle plague (Rinderpest) and drought
2. Population dispersal during some drought events was also due to social tensions/war
3. Low school attendance was at times due to the combined factors of drought and social tensions/war
4. Drought variably (directly or indirectly) caused social tensions and theft (i.e. as either a primary or secondary causative factor)
5. Indigenous rain making is only referred to during the 1865-69 drought in our documentary records - this does not imply that the practice was absent during other drought events









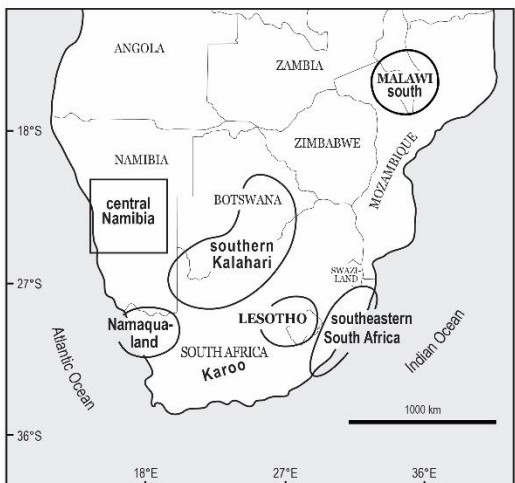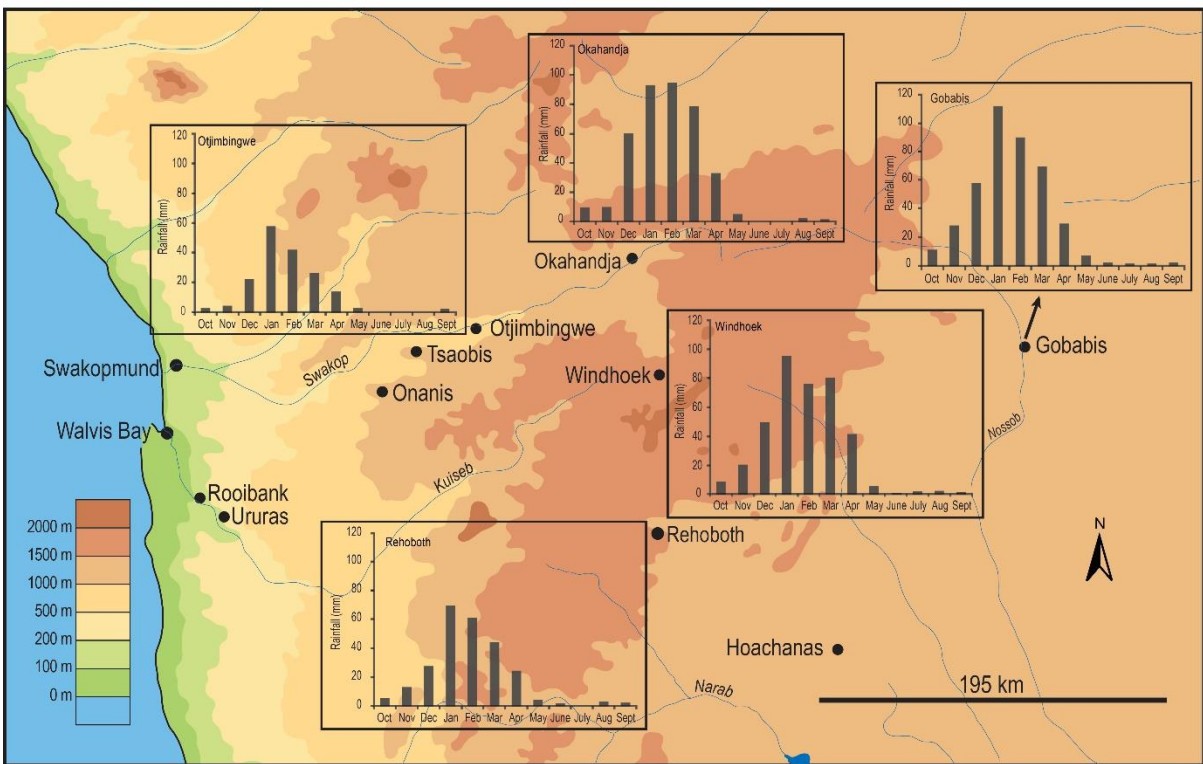


Figure 1:  The map of southern Africa indicates the central Namibia study region and other
areas for which documentary based 19[th] C climate reconstructions are available (please also
see Figure 4).    The topographic map of central Namibia indicates the location of primary
mission stations and their mean monthly rainfall during the period 1891-1913.


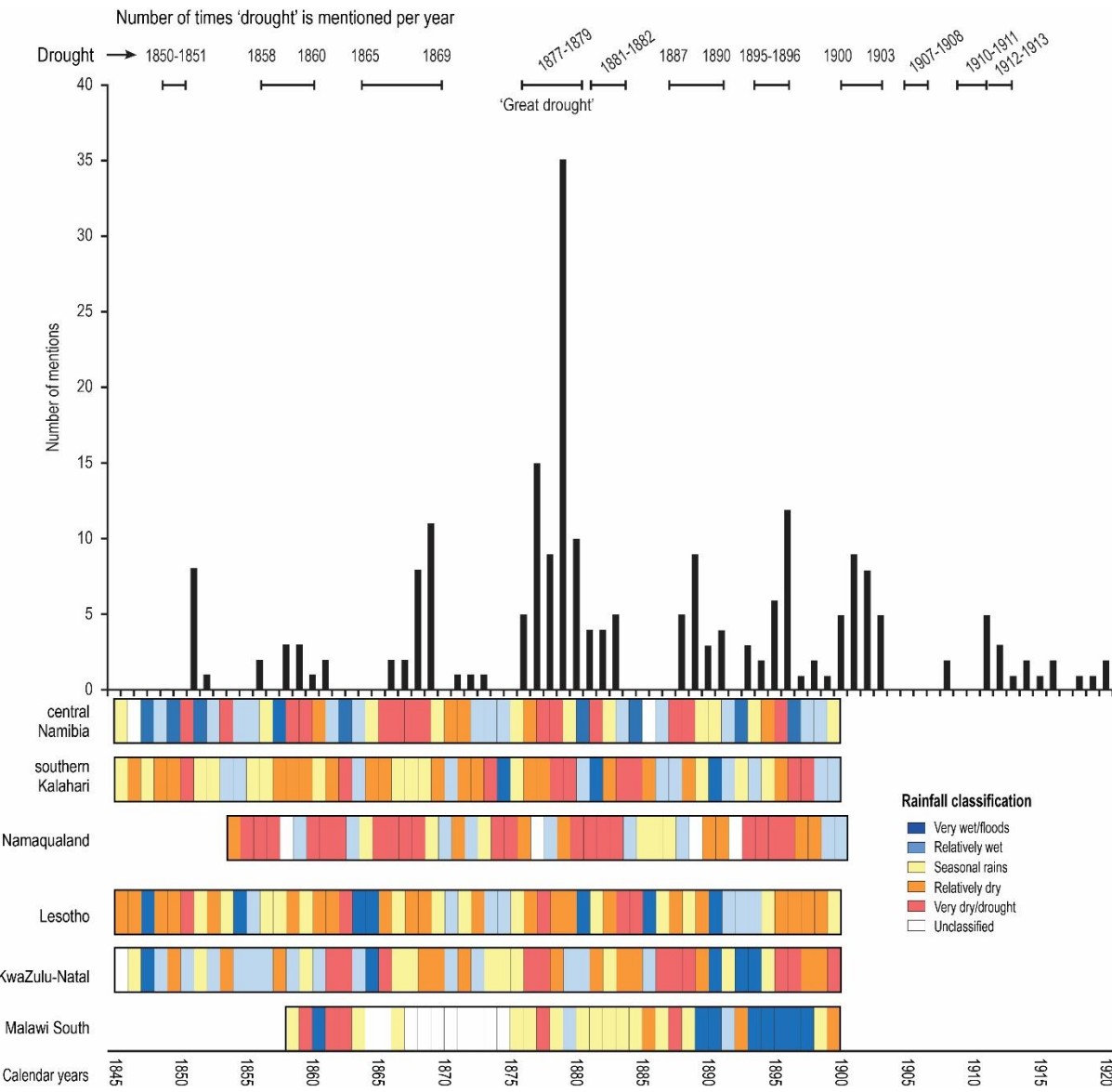



Figure 2: Annual 19th C rainfall reconstruction for southern African sub-regions (see also Figure 1). The bar graph indicates the number of times 'drought' is mentioned in central Namibian documentary sources each year (please note that these results are at least in part influenced by documentary source types and quantity).





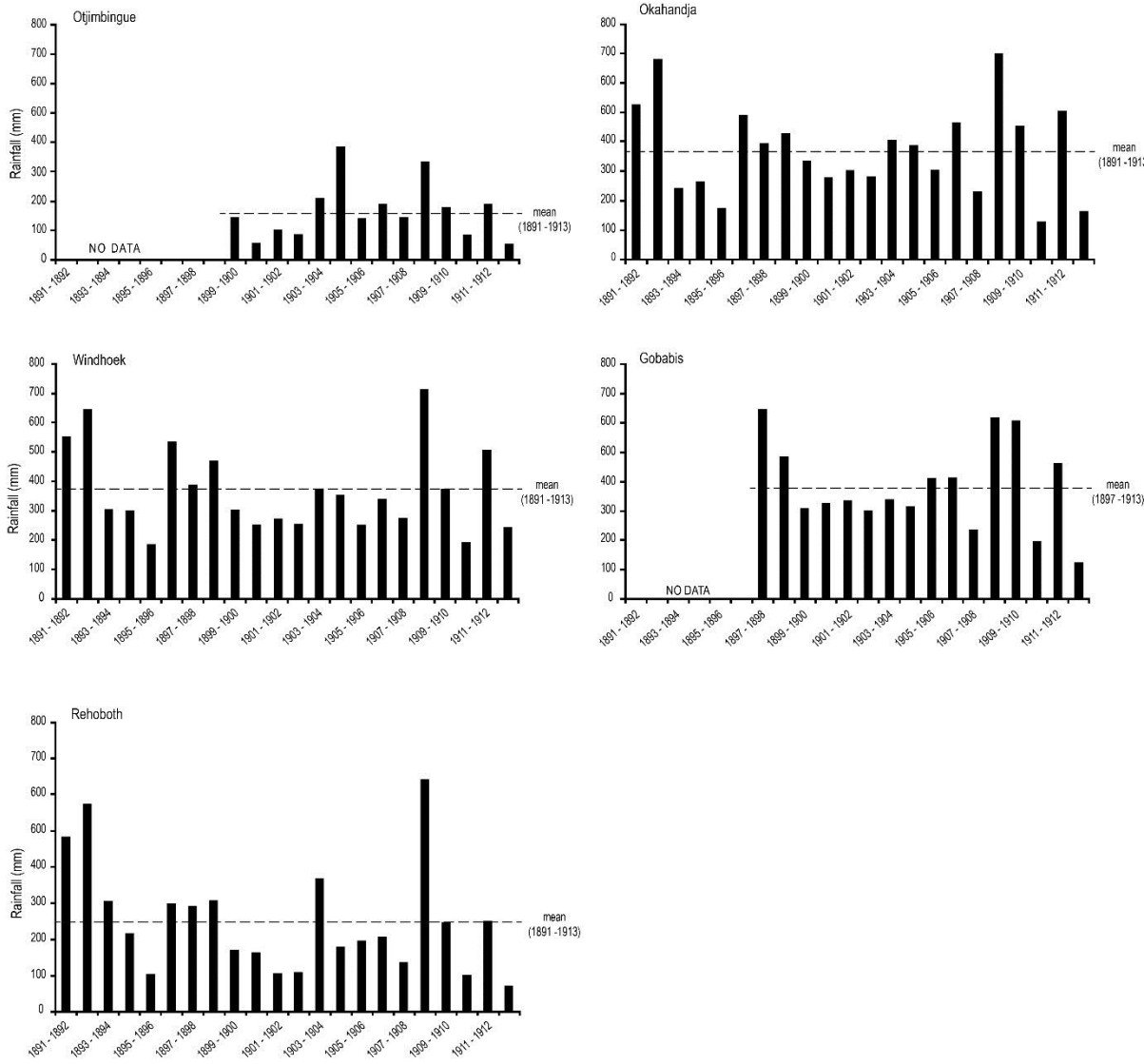



Figure 3: Wet season (Nov-April) rainfall totals for various stations between 1891 and 1913.







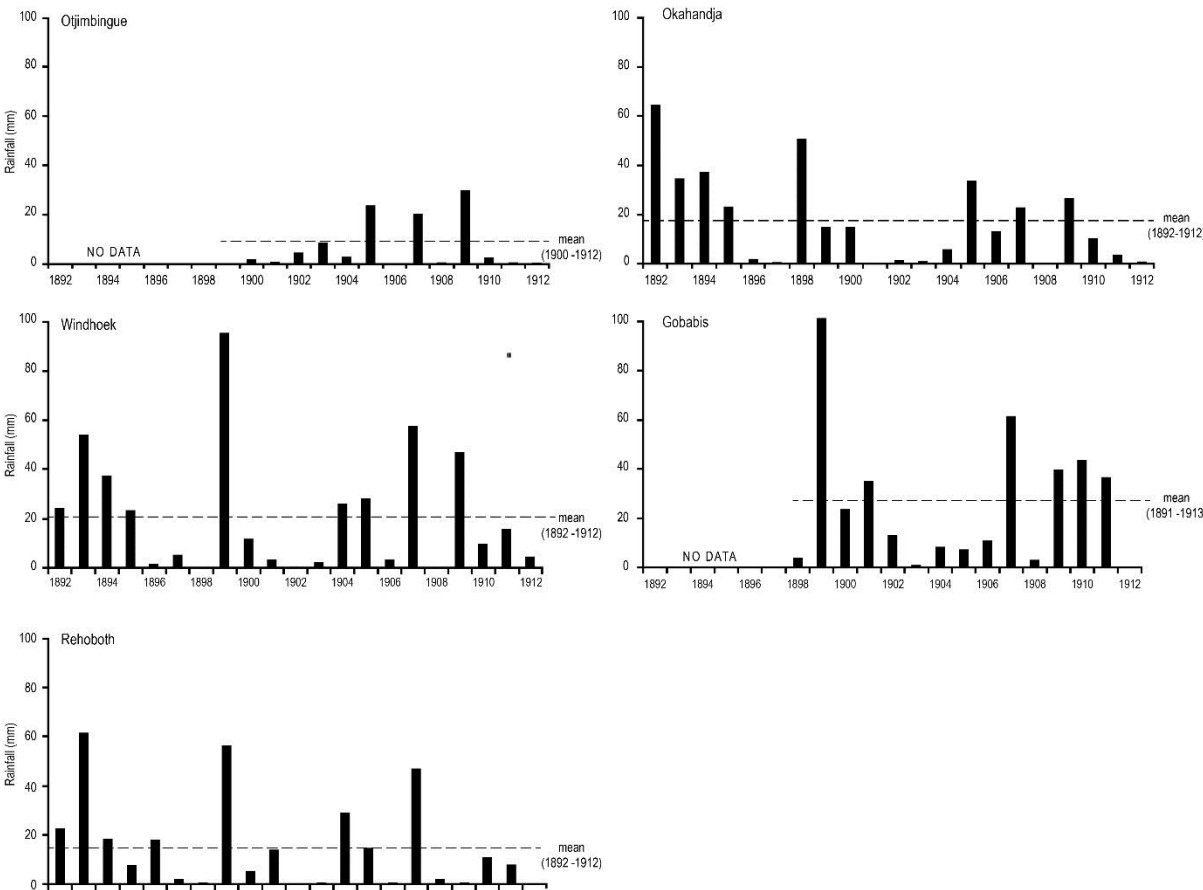



Figure 4: Dry season (May-Oct) rainfall totals for various stations between 1891 and 1913.
