# Peer review of ""Everything is scorched by the burning sun": Missionary perspectives and"

_Climate of the Past, 2019_

## Referee Comment (RC1) · Anonymous Referee #1 · 15 Oct 2019

Overall: This is a decent study that outlines the impact of major droughts in Namibia during the 19th and early 20th centuries. Usually I would think that a paper of this sort was an unusual one for CP, but it seems that it is intended for the forthcoming special issue on droughts, where I think it will be a good fit. The paper shows good potential, but needs a bit of work to get it up to scratch. In particular I have the following general comments: • The paper lacks a firm justification for what exactly a historical lens can bring. At the moment the justification is short, and somewhat contradictory. Adamson et al. (2018) in Global Environmental Change is helpful here, as well as similar work

on southern Africa by Kelso and Vogel (2015) also in GEC and Hannaford's recent papers in Global and Planetary Change and WIREs Climate Change • There is some confusion over the use of the word 'drought'. This is partly due to the use of the term drought in Grab and Zumthrum (2018) (which I would drop in favour of 'very dry'). The terms 'regional drought' and 'major drought' are also used without being explained, and I would like to see a little more description of which type of drought they are referring to at various places (agricultural, socioeconomic, ecological etc) • The discussion doesn't compare with any of the other similar studies that have been done in the region. In particular, Kelso and Vogel's work would make a very interesting comparison in terms of impacts, and Endfield and Nash's work in terms of the way that missionaries recorded drought.

Specific comments are below:

Abstract: spelling of achieved?

Lines 35-47 – the justification needs strengthening. It seems to suggest that the conditions under which ecological drought occur now are different to any in the past; i.e. they are related not to rainfall deficiencies (because of engineering) but to population growth. If this is the case, what exactly can historical case studies contribute? (To be clear I'm not saying they have no contribution to make – just that a better justification is required.)

Line 129-130 – where do these terms come from (climatic, consequential, social responsive, environmental) and what exactly do they mean?

Line 183-208 – the point is well taken, but how did you deal with it? Did you, for example, ignore any mention of the word 'drought' that came from missionaries who had been in country for less than three years? (This comes up again later.)

Section 3.2 – This section feels like it should come after section 3.3.

Line 221 – I assume the terms 'sufficient' and 'relatively wet' come from Grab and

Zumthurm (2018)? If so I would suggest reproducing the key figure(s) from this paper.

Line 222 – This term 'regional drought' needs more explanation. Is there a corresponding 'local' or 'widespread' drought? How does a regional drought relate to a 'major' drought, given later?

Lines 259-262 – A few things here. Firstly, the authors need to be clearer on what they mean by 'major drought'. Secondly, more detail is needed on how very dry years were assigned in Grab and Zumthrum (2018), and I would include at least one figure from this paper (see above). I would also stick with the term 'very dry' and drop '(drought)' as this is a little confusing and confounds the meteorological droughts assigned by the rainfall reconstruction with the other types of drought (agricultural, socioeconomic, ecological) of interest in this paper.

Line 262-265 – Is it suitable to just count the number of references to drought, given the issue with new missionaries into the area that were previously mentioned? I wonder if the authors should remove all mentions of the word from those who have been in the country for three years or less, given that this period is the length of time they suggest it was taking a missionary to understand what normal conditions were.

Section 3.4 – On the whole there could be a lot more information given about these droughts, if the word count allows

Section 4 – In general this is fine, but there is no comparison with any other studies. How does what was happening in Namibia compare with neighbouring regions? Kelso and Vogel's work on Namaqualand is particularly important here, as well as Endfield and Nash's work on the Kalahari.

---

## Referee Comment (RC2) · Anonymous Referee #2 · 21 Oct 2019

This article provides rich detail of the drought history and associated societal consequences of central Namibia from the mid-19th to early-20th century. The detail and challenges covered in this paper will undoubtedly be useful for historical climatologists engaged in drought reconstruction methods from colonial sources. It was also nice to see such a study crossing the somewhat artificial but very real dividing line from 19th to 20th century, which I think is an important step for African climate history.

One of my main overall comments, however, is that the paper is overly descriptive and leaves one wondering what are the meanings and implications beyond the context of

[Figure]

Namibia. This started to come out in the conclusion, but even then the conclusion that human experience and reporting of drought depends on social and environmental context is now fairly well-acknowledged in social scientific and humanities literature on climate. My main suggestion would therefore be to set up the paper with firmer and sharper research questions rather than the aim of simply establishing changes in influence and impact of drought over time, which could, in turn, provide for some sharper conclusions. As detailed in one of my comments below, one way this could be achieved is by grounding the paper in, and comparing it to, other missionary-derived climate histories in the region which cover changes in drought and its impacts over time - the prime example being the study by Kelso and Vogel (2015) in Global Environmental Change, but also studies by Nash and Endfield on the Kalahari, and Hannaford (2018) in Global and Planetary Change which takes an even longer view. In my view, this would be a more convincing way into the issues discussed in the paper rather than just the issue of drought definition and human engineering. It would also add something more to the growing regional body of work on historical drought-society interactions, for example by asking whether the Namibian case is unique, or whether we see similar patterns in impacts and perceptions as elsewhere (which section 4 in particular lends itself towards).

A number of specific comments are detailed below:

Line 46-47 - relating to the paragraph above in this review, what exactly are these lessons? The point about changing definitions and the conditions that can bring about 'drought' is noted, but what are the lessons that can be learnt from the past and what is the particular relevance of the before the era of human engineering?

Lines 128-130 - what do these characteristics mean? Presumably they are categorising the social-environmental characteristics of a drought, but the scale and characteristics of these categories are unclear.

Lines 142-181 - are these paragraphs 'results' as such? This is historical environmental

and social context.

Line 165 - 'Consequently, political and economic dominance was tangible' - this could do with some more explanation, i.e. how did the political intersect with the economic?

Lines 183-201 - this is a point that crops up in colonial accounts in many contexts and is an interesting one. It would be valuable to know how the authors dealt with this issue for 'newcomers' to Namibia; was the word 'drought' simply discounted for these observers?

Lines 259-260 - I would suggest considering reformulating this sentence (it sounds a bit more like an email than a scientific journal paper!). It would also be useful to provide some more material from Grab and Zumthurm (2018) (e.g. the drought classifications and chronology), which seems to be of fundamental importance to this article.

Lines 262-266 - Table 1 is a really nice visualisation of drought impacts. However, there are some issues with 'drought mentions' as a proxy for drought occurrence, if this is the intention of the figure. The authors do acknowledge that this can be dictated by the availability of documentary material, but there may also be other issues here, e.g. the length of time a missionary had been resident in Namibia. There is also the issue of the extent of alignment between Table 1 and Figure 4, e.g. the drought of 1877-1879 had most of the 'reported consequences' categories ticked whilst also being the drought that was most mentioned, which one might expect, but this was closely followed in breadth of reported consequences by the drought of 1900-1903, yet the discrepancy in drought mentions is very large indeed. Why is this?

Section 4 - this section provides a nice social-environmental chronology and is rich in detail. It relates this chronology to the larger southern African picture, though only in terms of drought periodisation rather than that of societal responses. It would be very valuable to see some comparative elements to this section, the most obvious example being the work by Kelso and Vogel (2015) on Namaqualand, which has a very similar temporal scope and would provide a fascinating comparison.

---

## Short Comment (SC1) · 11 Nov 2019

Dear Anonymous Reviewer

I really value your comments. One that in particular struck me is the questioning about differentiating between 'regional drought' and 'major drought' given that these have not been adequately defined. In reflecting on this, I think drought terminology needs more careful consideration in documentary based climate work.

I have just had a look through several prominent documentary based climate recon-
struction papers (and some already published in this special volume on droughts) and many, if not most, use drought terms rather vaguely. The irony is that the discipline has been generally good in classifying - through defining attributes - different categories of wetness or dryness, but when it comes to 'drought' - we have generally failed in this regard. So, papers have used terms such as 'drought', 'moderate drought' and 'severe drought' all in the same paper without any differentiation between these categories of drought. Other papers have used terms such as 'major drought', 'extreme drought', 'very great drought', 'extraordinary drought' etc...all without any clarification how these might differ from 'drought'. The work by Tejedor et al and Brazdil et al (both this volume) - are exceptions to this as they quantify and explain, respectively, what constitutes an 'extreme drought'. Most other papers have failed to do so - including our own here - so thanks for highlighting this to us.

Defining the category of drought I guess depends on ones frame of reference and geographic work space etc - so what might be a 'severe' drought to one region may only qualify as a 'moderate' one to another, or may in fact not even qualify as a drought if in a much drier region. So collectively we have much to improve on defining categories of drought in terms of their severity, extent, duration etc.

I will make sure we cover ourselves better in this regard when revising the current paper. Many thanks again for this interesting and important point. Stefan

---

## Author Comment (AC1) · 2 Dec 2019

Anonymous Referee #1 Overall: This is a decent study that outlines the impact of major droughts in Namibia during the 19th and early 20th centuries. Usually I would think that a paper of this sort was an unusual one for CP, but it seems that it is intended for the forthcoming special issue on droughts, where I think it will be a good fit.

YES, THE PAPER IS FRAMED TO SUITE THE SPECIAL ISSUE ON HISTORICAL

DROUGHTS AND HAVING SEEN SEVERAL OF THE OTHER SUBMISISONS MADE WE FEEL THAT OUR PAPER IS VERY MUCH SUITABLE TO THIS SPECIAL IS- SUE AND IS PARTICULARLY VALUABLE GIVEN THAT IT IS THE ONLY CONTRI- BUTION TO AFRICA AND ALSO REPRESENTS A CONTRIBUTION CONCERNING DROUGHTS FROM A SEMI-ARID REGIONS AND THUS ADD NEW PERSPECTIVES ON DROUGHT.

The paper shows good potential, but needs a bit of work to get it up to scratch. In particular I have the following general comments: The paper lacks a firm justification for what exactly a historical lens can bring. At the moment the justification is short, and somewhat contradictory. Adamson et al. (2018) in Global Environmental Change is helpful here, as well as similar work on southern Africa by Kelso and Vogel (2015) also in GEC and Hannaford's recent papers in Global and Planetary Change and WIREs Climate Change .

YES, THIS POINT IS WELL TAKEN. WE HAVE REMOVED THE TEXT 'HISTORICAL LENS' AS IT CREATED CONFUSION WHAT THIS MIGHT BE. WE HAVE EXPANDED THE TEXT IN TWO PARTS OF THE INTRODUCTION TO ADDRESS THESE CON- CERNS. IN PARTICULAR, WE HAVE NOW EXPANDED THE LAST SECTION OF THE INTRODUCTION TO MAKE IT MUCH CLEARER AS TO WHAT EXACTLY THIS PAPER AIMS TO DO. For this reason, there is value in exploring drought contexts through a window of time when the natural-human environment was rapidly trans- formed into a more human-engineered environment (through colonial conquests). For instance, it may provide insight to how drought impacted past indigenous populations and the environment, in ways that may no longer apply today, such as water-resource contexts during periods of nomadic lifestyles. AND This then provides us with an oppor- tunity to establish similarities and differences in 19th C drought-related circumstances and experiences through dryland regions of southern Africa. More particularly, we aim to:1) outline the historic context of meteorological/hydrological drought over cen- tral Namibia, 2) provide evidence for the (at times) relatively complex geographic nature (spatial/temporal) of such droughts in the region, 3) summarize central Namibian drought events between 1850 and 1920, and 4) establish the temporal shifts of influence and impact that historical droughts had on society and the environment during this period, as portrayed in written documents.

There is some confusion over the use of the word 'drought'. This is partly due to the use of the term drought in Grab and Zumthurm (2018) (which I would drop in favour of 'very dry'). The terms 'regional drought' and 'major drought' are also used without being explained, and I would like to see a little more description of which type of drought they are referring to at various places (agricultural, socioeconomic, ecological etc).

TO US, THIS IS A VERY INTERESTING COMMENT AND ONE WE HAVE REFLECTED ON CONSIDERABLY. WE ENSURE TO EXPLAIN MORE PRECISELY THAT WHAT WE ARE DEALING WITH ARE HYDRO/METEOROLOGICAL DROUGHTS. WE HAVE ALSO ELABORATED ON THIS MATTER IN THE INTERACTIVCE DISCUSSION. I HEREBY ATTACH THE COMMENTS IN ITALICS: Dear Anonymous Reviewer I really value your comments. One that in particular struck me is the questioning about differentiating between 'regional drought' and 'major drought' given that these have not been adequately defined. In reflecting on this, I think drought terminology needs more careful consideration in documentary based climate work. I have just had a look through several prominent documentary based climate reconstruction papers (and some already published in this special volume on droughts) and many, if not most, use drought terms rather vaguely. The irony is that the discipline has been generally good in classifying - through defining attributes - different categories of wetness or dryness, but when it comes to 'drought' - we have generally failed in this regard. So, papers have used terms such as 'drought', 'moderate drought' and 'severe drought' all in the same paper without any differentiation between these categories of drought. Other papers have used terms such as 'major drought', 'extreme drought', 'very great drought', 'extraordinary drought' etc...all without any clarification how these might differ from a 'normal' drought. The work by Tejedor et al and Brazdil et al (both this volume)

- are exceptions to this as they quantify and explain, respectively, what constitutes an 'extreme drought'. Most other papers have failed to do so - including our own here – so thanks for highlighting this to us. Defining the category of drought I guess depends on ones frame of reference and geographic work space etc - so what might be a 'severe' drought to one region may only qualify as a 'moderate' one to another, or may in fact not even qualify as a drought if in a much drier region. So collectively we have much to improve on defining categories of drought in terms of their severity, extent, duration etc. I will make sure we cover ourselves better in this regard when revising the current paper. Many thanks again for this interesting and important point. Stefan

WE WOULD, HOWEVER, DISAGREE WITH THE REVIEWER'S SUGGESTION THAT WE CHANGE THE TERM 'DROUGHT' TO 'VERY DRY' BECAUSE ALL 'VERY DRY' SITUATIONS DISCUSSED ARE IN OUR OPINION SYNONYMOUS TO 'DROUGHT' AND ALL YEARS CLASSIFIED AS 'VERY DRY' ARE IN DROUGHT. ALSO, WE PRE-FER TO USE THE TERM 'DROUGHT' TO THE TERM 'VERY DRY' BECAUSE IT COMPLIES WITH THE LITERATURE ON DROUGHT. IF WE WERE TO CHANGE WHAT REALLY IS A 'DROUGHT' TO RE-DEFINING IT AS 'VERY DRY' ,THEN THIS WOULD NO LONGER COMPLY WITH THE LITERATURE AND ALSO NO LONGER COMPLY WITH ANY OF THE OTHER PAPERS IN THIS SPECIAL ISSUE AS THE TERMINOLOGY WOULD DIFFER. WE DO, HOWEVER, MAKE IT VERY CLEAR FROM THE START WHAT WE UNDERSTAND DROUGHT TO BE FOR OUR PAPER – HERE IS AN EXTRACT OF OUR REVISED INTRODUCTION: In this special issue, Brázdil et al. (2019) explore various types and characteristics of drought that are relevant to both contemporary and historical contexts. These authors use the definition by Wilhite and Pulworty (2018) to define drought as 'a prolonged period of negative deviation in water balance compared to the climatological norm in a given area' (p1915). Although quantification of 'cimatological norms' during pre-instrumental periods is challenging, if at all possible, we broadly follow Wilhite and Pulworty's definition of drought for our current work.

The discussion doesn't compare with any of the other similar studies that have been done in the region. In particular, Kelso and Vogel's work would make a very interesting comparison in terms of impacts, and Endfield and Nash's work in terms of the way that missionaries recorded drought.

ALTHOUGH WE WERE FAMILIAR WITH THESE PAPERS, WE HAVE NOW READ ALL THESE PAPERS AGAIN TO ADDRESS THIS CONCERN. ON HAVING READ THEM AGAIN, WE REALIZE JUST HOW DIFFERENT (AND IN OUR VIEW WE BE-LIEVE 'UNIQUE') OUR PAPER IS, IN TERMS OF WHAT IT PRESENTS CONCERN-ING HISTORICAL DROUGHTS IN SOUTHERN AFRICA. OUR PAPER DEMON-STRATES SOME IDENTIFIED TEMPORAL CONSEQUENTIAL AND HUMAN RE-SPONSIVE PATTERNS TO DROUGHT, WHICH NONE OF THESE OTHER PAPERS ADDRESS. ALTHOUGH ALL THE WORK FROM THESE OTHER REGIONS IS EX-CEPTIONALLY INTERESTING AND VALUABLE TO US, THESE OTHER PUBLISHED WORKS ARE NOT DIRECTLY COMPARABLE TO WHAT WE PRESENT. IN FACT IT IS DIFFICULT TO MAKE ANY STRONG COMPARISONS WITH OUR PAPER BE-CAUSE THE WAY THESE OTHER PAPERS ARE THEORETICALLY FRAMED - ALL RATHER DIFFERENT TO THE WAY OUR PAPER IS FRAMED. HOWEVER, WE HAVE LOOKED VERY CAREFULLY WHERE WE MIGHT BE ABLE TO CITE THESE PAPERS WHERE THERE IS SOME RELEVANCE TO WHAT WE DISCUSS. THE STRONGEST LINKS WE FOUND WERE IN THE KELSO AND VOGEL (2015) WORK WHERE THESE AUTHORS ADDRESS DROUGHT AND RESILIENCE THROUGH THE 19THC IN NAMAQUALAND AND WHERE WE ARE INDEED ABLE TO MAKE SOME RELEVANT LINKS – WHICH WILL NOW BE MADE AS WE REVISE THE MANUSCRIPT. BUT EVEN HERE, MANY OF THE THINGS WE DISCUSS CON-CERNING WATER AND THE ENVIRONMENT ETC ARE NOT DISCUSSION POINTS FOR NAMAQUALAND AND OTHER SUB-REGIONS OF SOUTHERN AFRICA (I.E. KALAHARI ETC).

Specific comments are below: Abstract: spelling of achieved? NOTED

Lines 35-47 – the justification needs strengthening. It seems to suggest that the conditions under which ecological drought occur now are different to any in the past; i.e. they are related not to rainfall deficiencies (because of engineering) but to population growth. If this is the case, what exactly can historical case studies contribute? (To be clear I'm not saying they have no contribution to make – just that a better justification is required.) THIS HAS NOW BEEN ELABORATED

Line 129-130 – where do these terms come from (climatic, consequential, social responsive, environmental) and what exactly do they mean? THIS HAS BEEN RE-WORKED NOW

Line 183-208 – the point is well taken, but how did you deal with it? Did you, for example, ignore any mention of the word 'drought' that came from missionaries who had been in country for less than three years? (This comes up again later.) WE EXPLAIN THAT THIS IS NOT A CONCERN GIVEN TRIANGULATION OF SOURCE INFORMATION AND CAREFUL SCRUTINY OF SUPPORTIVE EVIDENCE

Section 3.2 – This section feels like it should come after section 3.3. YES, HAVE CHANGED THIS AS REQUESTED

Line 221 – I assume the terms 'sufficient' and 'relatively wet' come from Grab and Zumthurm (2018)? If so I would suggest reproducing the key figure(s) from this paper. 0K, WE CAN INCLUDE ANOTHER FIGURE TO COVER FOR THIS

Line 222 – This term 'regional drought' needs more explanation. Is there a corresponding 'local' or 'widespread' drought? How does a regional drought relate to a 'major' drought, given later? YES, WE CAN ADDRESS THESE TERMINOLOGICAL CONCERNS ACCORDINGLY AND IN FACT REMOVE THESE SUB-CATEGORIES OF DROUGHT TO AVOID SUCH CONCERNS.

Lines 259-262 – A few things here. Firstly, the authors need to be clearer on what they mean by 'major drought'. Secondly, more detail is needed on how very dry years were

assigned in Grab and Zumthrum (2018), and I would include at least one figure from this paper (see above). I would also stick with the term 'very dry' and drop '(drought)' as this is a little confusing and confounds the meteorological droughts assigned by the rainfall reconstruction with the other types of drought (agricultural, socioeconomic, ecological) of interest in this paper. YES, THESE CONCERNS ARE SIMILAR TO THOSE ADDRESSED ABOVE AND TO WHICH WE HAVE RESPONDED ABOVE. BUT YES, WE WILL ADDRESS THE TERMINOLOGY ISSUES AND ADD THAT EXTRA FIGURE.

Line 262-265 – Is it suitable to just count the number of references to drought, given the issue with new missionaries into the area that were previously mentioned? I wonder if the authors should remove all mentions of the word from those who have been in the country for three years or less, given that this period is the length of time they suggest it was taking a missionary to understand what normal conditions were. WE WILL ELABORATE WHAT WE INTEND TO ACHIEVE WITH THIS FIGURE IN THE FIGURE CAPTION. THE FIGURE IS NOT INTENDED TO INFORM WHEN THE DROUGHTS OCCURRED BECAUSE OUR IDENTIFICATION OF PERIODS OF DROUGHT IS NOT BASED ON NUMBER OF MENTIONS OF DROUGHT AT ALL. IN FACT WE WISH TO SHOW JUST THE OPPOSITE, NAMELY THAT WHEN WE HAD PERIODS OF IDENTIFIED DROUGHT, THERE WAS MUCH MORE WRITTEN ABOUT DROUGHT IN VARIOUS DOCUMENTS AND SO THE FIGURE SERVES TO SHOW JUST HOW OFTEN DROUGHT IS MENTIONED BY YEAR….AND IN FACT IT IS MENTIONED IN SOME YEARS WHEN WE DO NOT ACTUALLY HAVE A DROUGHT. BUT WE WILL EXPLAIN THIS BETTER IN THE TEXT.

Section 3.4 – On the whole there could be a lot more information given about these droughts, if the word count allows WE DID NOT WISH TO ELABORATE FURTHER ON THESE DROUGHTS BECAUSE OTHER DETAILS ABOUT THE SAME DROUGHT ARE ALREADY ELABORATED ON IN GRAB & ZUMTHURM (2018). IT WOULD NOT BE ETHICAL FOR US TO DUPLICATE THAT WHICH HAS ALERADY BEEN

PUBLISHED AND SO WE WERE VERY CAREFULL TO JUST PROVIDE A SUM-
MARY OF THE DROUGHTS AND ASPECTS OF THE DROUGHTS THAT HAVE NOT
BEEN METIONED IN THE PREVIOUS PAPER OF OURS. HOWEVER, WE AGREE
THAT THE SECTION COULD BE EXPANDED AND GAVE THIS SOME FURTHER
THOUGHT AS TO WHAT MIGHT BE MOST APPROPRIATE HERE. WE FEEL THAT
THE BEST WAY TO EXPAND THE SECTION MAY BE TO COMAPRE OUR CENTRAL
NAMIBIA DROUGHT PERIODS WITH CONDITIONS OVER THE ADJOINING KALA-
HARI AND NAMAQUA REGIONS. THIS WOULD ALSO THEN ADDERSS SOME
EARLIER CONCERNS THAT WORK DONE IN THESE ADJOINING REGIONS DE-
SERVE A BIT MORE REFERENCE.

Section 4 – In general this is fine, but there is no comparison with any other studies.
How does what was happening in Namibia compare with neighbouring regions? Kelso
and Vogel's work on Namaqualand is particularly important here, as well as Endfield
and Nash's work on the Kalahari. WHERE RELEVANT, WE NOW MAKE REFERENCE
TO SUCH WORK.

---

## Author Comment (AC2) · 2 Dec 2019

Anonymous Referee #2 This article provides rich detail of the drought history and associated societal consequences of central Namibia from the mid-19th to early-20th century. The detail and challenges covered in this paper will undoubtedly be useful for historical climatologists engaged in drought reconstruction methods from colonial sources. It was also nice to see such a study crossing the somewhat artificial but very real dividing line from 19th to 20th century, which I think is an important step for African climate history. One of my main overall

comments, however, is that the paper is overly descriptive and leaves one wondering what are the meanings and implications beyond the context of Namibia.

THE NATURE OF DOCUMENTARY BASED WORK IS IN ESSENCE OFTEN VERY DESCRIPTIVE, AS IS THE CASE WITH OTHER PAPERS IN THIS SPECIAL ISSUE ON DROUGHTS. DESCRIPTIVE APPROACHES CAN BE RICH IN DETAIL, OFTEN SHARING MORE INSIGHT THAN NON-DESCRIPTIVE APPROACHES. HOWEVER, HAVING SAID THIS, WE WOULD ARGUE THAT THERE ARE IN FACT NUMER-ICAL (NON-DESCRIPTIVE) APPROACHES THAT WE HAVE WEAVED INTO THE MANUSCRIPT TO PROVIDE A BALANCED APPROACH.

This started to come out in the conclusion, but even then the conclusion that human experience and reporting of drought depends on social and environmental context is now fairly well-acknowledged in social scientific and humanities literature on climate. My main suggestion would therefore be to set up the paper with firmer and sharper research questions rather than the aim of simply establishing changes in influence and impact of drought over time, which could, in turn, provide for some sharper con-clusions. WE HAVE NOW EXPANDED THE LAST SECTION OF THE INTRODUC-TION TO MAKE IT MUCH CLEARER AS TO WHAT EXACTLY THIS PAPER AIMS TO DO. SO IT SETS A MUCH FIRMER SET OF AIMS, TO WHICH THE CONCLUSION SPEAKS.

As detailed in one of my comments below, one way this could be achieved is by ground-ing the paper in, and comparing it to, other missionary-derived climate histories in the region which cover changes in drought and its impacts over time - the prime exam-ple being the study by Kelso and Vogel (2015) in Global Environmental Change, but also studies by Nash and Endfield on the Kalahari, and Hannaford (2018) in Global and Planetary Change which takes an even longer view. In my view, this would be a more convincing way into the issues discussed in the paper rather than just the issue of drought definition and human engineering. It would also add something more to the growing regional body of work on historical drought-society interactions, for example

by asking whether the Namibian case is unique, or whether we see similar patterns in impacts and perceptions as elsewhere (which section 4 in particular lends itself towards).

ALTHOUGH WE WERE FAMILIAR WITH THESE PAPERS, WE HAVE NOW READ ALL THESE PAPERS AGAIN TO ADDRESS THIS CONCERN. ON HAVING READ THEM AGAIN, WE REALIZE JUST HOW DIFFERENT (AND IN OUR VIEW WE BELIEVE 'UNIQUE') OUR PAPER IS, IN TERMS OF WHAT IT PRESENTS CONCERNING HISTORICAL DROUGHTS IN SOUTHERN AFRICA. OUR PAPER DEMONSTRATES SOME IDENTIFIED TEMPORAL CONSEQUENTIAL AND HUMAN RESPONSIVE PATTERNS TO DROUGHT, WHICH NONE OF THESE OTHER PAPERS ADDRESS. ALTHOUGH ALL THE WORK FROM THESE OTHER REGIONS IS EXCEPTIONALLY INTERESTING AND VALUABLE TO US, THESE OTHER PUBLISHED WORKS ARE NOT DIRECTLY COMPARABLE TO WHAT WE PRESENT. IN FACT IT IS DIFFICULT TO MAKE ANY STRONG COMPARISONS WITH OUR PAPER BECAUSE THE WAY THESE OTHER PAPERS ARE THEORETICALLY FRAMED - ALL RATHER DIFFERENT TO THE WAY OUR PAPER IS FRAMED. HOWEVER, WE HAVE LOOKED VERY CAREFULLY WHERE WE MIGHT BE ABLE TO CITE THESE PAPERS WHERE THERE IS SOME RELEVANCE TO WHAT WE DISCUSS. THE STRONGEST LINKS WE FOUND WERE IN THE KELSO AND VOGEL (2015) WORK WHERE THESE AUTHORS ADDRESS DROUGHT AND RESILIENCE THROUGH THE 19THC IN NAMAQUALAND AND WHERE WE ARE INDEED ABLE TO MAKE SOME RELEVANT LINKS – WHICH WILL NOW BE MADE AS WE REVISE THE MANUSCRIPT. BUT EVEN HERE, MANY OF THE THINGS WE DISCUSS CONCERNING WATER AND THE ENVIRONMENT ETC ARE NOT DISCUSSION POINTS FOR NAMAQUALAND AND OTHER SUB-REGIONS OF SOUTHERN AFRICA (I.E. KALAHARI ETC).

A number of specific comments are detailed below: Line 46-47 - relating to the paragraph above in this review, what exactly are these lessons? The point about changing

definitions and the conditions that can bring about 'drought' is noted, but what are the lessons that can be learnt from the past and what is the particular relevance of the before the era of human engineering? WE HAVE REWORKED THE TEXT TO ADDRESS THESE ISSUES

Lines 128-130 - what do these characteristics mean? Presumably they are categorising the social-environmental characteristics of a drought, but the scale and characteristics of these categories are unclear. WE WILL BE CLEARER WITH THIS

Lines 142-181 - are these paragraphs 'results' as such? This is historical environmental and social context.

YES- THEY ARE RESULTS

Line 165 - 'Consequently, political and economic dominance was tangible' - this could do with some more explanation, i.e. how did the political intersect with the economic? WE HAVE ELABORATED AS FOLLOWS: Consequently, political and economic dominance was tangible. In particular, much of central Namibia's economy functioned through cattle, which was viewed to be the best option to store wealth, as it was easily transferable. Combined with smart and shifting alliance-making, large herds of cattle allowed its controller to enforce tribute-systems or to claim land and thus ensure political dominance. Such a socio-economic system was, however, easily disrupted through a variety of factors such as drought, conflict, cattle diseases and European colonization/influence. Ultimately, such an indigenous socio-economy gradually declined in significance as European influences rapidly increased through the late 19th/early 20th centuries.

Lines 183-201 - this is a point that crops up in colonial accounts in many contexts and is an interesting one. It would be valuable to know how the authors dealt with this issue for 'newcomers' to Namibia; was the word 'drought' simply discounted for these observers? WE EXPLAIN THAT THIS IS NOT A CONCERN GIVEN TRIANGULATION OF SOURCE INFORMATION AND CAREFUL SCRUTINY OF SUPPORTIVE

EVIDENCE AS FOLLOWS: To this end, and where possible, comments on weather, climate and the environment require careful scrutiny and comparison across various sources. In most cases written texts contain valuable contextual information (e.g. dryness/wetness of river channels, poor state of shrubs and trees, comments from older indigenous inhabitants etc) which helps verify claims of drought. In addition, several missionaries resided and travelled extensively in central Namibia for many years and in some instances decades (e.g. Viehe: 26yrs; Hahn: 30yrs; Heidmann: 39yrs; Bernsmann: 42yrs; Irle: 47yrs; Diehl: 51yrs), constantly interacting with local community members. In such cases, missionaries developed excellent knowledge of the local weather patterns and climate, and were able to place contemporary climatic conditions in perspective, comparing situations with those experienced over many years prior.

Lines 259-260 - I would suggest considering reformulating this sentence (it sounds a bit more like an email than a scientific journal paper!). It would also be useful to provide some more material from Grab and Zumthurm (2018) (e.g. the drought classifications and chronology), which seems to be of fundamental importance to this article. SURE – WE CAN REWORK THIS

Lines 262-266 - Table 1 is a really nice visualisation of drought impacts. However, there are some issues with 'drought mentions' as a proxy for drought occurrence, if this is the intention of the figure. The authors do acknowledge that this can be dictated by the availability of documentary material, but there may also be other issues here, e.g. the length of time a missionary had been resident in Namibia. There is also the issue of the extent of alignment between Table 1 and Figure 4, e.g. the drought of 1877-1879 had most of the 'reported consequences' categories ticked whilst also being the drought that was most mentioned, which one might expect, but this was closely followed in breadth of reported consequences by the drought of 1900-1903, yet the discrepancy in drought mentions is very large indeed. Why is this?

YES – WE HAVE ADDRESSED THIS AND EXPANDED THE TEXT CONSIDERABLY TO EXPLAIN ALL THIS MORE FULLY TO AVOID SUCH CONCERNS BY THE

READER: Figure 4 lists the number of times 'drought' is mentioned in documentary sources each year, and how this compares with the hydro-meteorological 19th C chronology by Grab and Zumthurm (2018). While the depicted results are impacted by documentary data availability and do not necessarily indicate drought severity, the intention with this figure is to provide a visual impression highlighting times when 'drought' received much mention (and thus attention) through written sources, such as during the significant drought events of 1865-69, 1877-79, 1895-96 and 1900-03. Although the 1900-1903 event does not receive as much mention (according to Figure 4) as those during 1895-96 and 1877-79, this is largely due to fewer documentary source materials having been consulted for times since ∼1900. The more recent documents contain a much greater detail of information, hence requiring fewer sources. However, the figure also demonstrates that concerns of drought conditions are reported much more frequently (66% of years) than the actual occurrence of drought (29% of years) during the 19th C. This is owing largely to conditions of [prolonged] seasonal aridity, usually described as 'drought'.

Section 4 - this section provides a nice social-environmental chronology and is rich in detail. It relates this chronology to the larger southern African picture, though only in terms of drought periodisation rather than that of societal responses. It would be very valuable to see some comparative elements to this section, the most obvious example being the work by Kelso and Vogel (2015) on Namaqualand, which has a very similar temporal scope and would provide a fascinating comparison. ALTHOUGH WE WERE FAMILIAR WITH THESE PAPERS, WE HAVE NOW READ ALL THESE PAPERS AGAIN TO ADDRESS THIS CONCERN. ON HAVING READ THEM AGAIN, WE REALIZE JUST HOW DIFFERENT (AND IN OUR VIEW WE BELIEVE 'UNIQUE') OUR PAPER IS, IN TERMS OF WHAT IT PRESENTS CONCERNING HISTORICAL DROUGHTS IN SOUTHERN AFRICA. OUR PAPER DEMONSTRATES SOME IDENTIFIED TEMPORAL CONSEQUENTIAL AND HUMAN RESPONSIVE PATTERNS TO DROUGHT, WHICH NONE OF THESE OTHER PAPERS ADDRESS. ALTHOUGH ALL THE WORK FROM THESE OTHER REGIONS IS EXCEPTIONALLY INTERESTING

AND VALUABLE TO US, THESE OTHER PUBLISHED WORKS ARE NOT DIRECTLY COMPARABLE TO WHAT WE PRESENT. IN FACT IT IS DIFFICULT TO MAKE ANY STRONG COMPARISONS WITH OUR PAPER BECAUSE THE WAY THESE OTHER PAPERS ARE THEORETICALLY FRAMED - ALL RATHER DIFFERENT TO THE WAY OUR PAPER IS FRAMED. HOWEVER, WE HAVE LOOKED VERY CAREFULLY WHERE WE MIGHT BE ABLE TO CITE THESE PAPERS WHERE THERE IS SOME RELEVANCE TO WHAT WE DISCUSS. THE STRONGEST LINKS WE FOUND WERE IN THE KELSO AND VOGEL (2015) WORK WHERE THESE AUTHORS ADDRESS DROUGHT AND RESILIENCE THROUGH THE 19THC IN NAMAQUALAND AND WHERE WE ARE INDEED ABLE TO MAKE SOME RELEVANT LINKS – WHICH WILL NOW BE MADE AS WE REVISE THE MANUSCRIPT. BUT EVEN HERE, MANY OF THE THINGS WE DISCUSS CONCERNING WATER AND THE ENVIRONMENT ETC ARE NOT DISCUSSION POINTS FOR NAMAQUALAND AND OTHER SUB-REGIONS OF SOUTHERN AFRICA (I.E. KALAHARI ETC).

---

## Author Response (AR2)

[revised manuscript text omitted]

2019).

Our paper makes an important contribution to the study of historical droughts, both globally
and more specifically to southern Africa (see Brázdil et al., 2018), by demonstrating the
imperative to evaluate historical drought events, not only according to meteorological
parameters, but also in consideration of changing natural-environmental and human-
environmental contexts through time. For this, written-documentary sources are an essential
and invaluable proxy record that ought to be more regularly considered when evaluating the
severity of past droughts.

**Acknowledgements**

We thank two anonymous referees and editor who provided valuable suggestions to help
improve the manuscript.

[revised manuscript text omitted]